# SSA: Sparse Sparse Attention
# by Aligning Full and Sparse Attention Outputs in Feature Space

Zhenyi Shen [* † 1 2]   Junru Lu [* 2]   Lin Gui [1]   Jiazheng Li [1]   Yulan He [1]   Di Yin [2]   Xing Sun [2]

## Abstract

Sparse attention reduces the quadratic complexity of full self-attention but faces two challenges: (1) an *attention gap*, where applying sparse attention to full-attention-trained models causes performance degradation due to train-inference distribution mismatch, and (2) a *capability gap*, where models trained purely with sparse attention lack complete gradient flow, preventing them from matching full-attention performance. We propose **SSA** (Sparse Sparse Attention), a training framework that integrates both sparse and full attention with bidirectional attention-output alignment. We prove that the approximation error scales linearly with the attention mass dropped under sparse attention, and show that SSA's alignment objective substantially reduces this quantity compared to baselines. Experiments demonstrate that SSA achieves state-of-the-art performance under both inference modes, adapts smoothly to varying sparsity budgets, and demonstrates superior long-context capabilities. Code is available at https://github.com/zhenyi4/ssa.

## 1. Introduction

With the blooming of large language models (LLMs), the demand for efficient long-context processing has grown substantially across diverse scenarios, including long-document understanding (Zhang et al., 2023; Jimenez et al., 2024), extended reasoning trajectories (OpenAI et al., 2024; DeepSeek-AI et al., 2025), and agentic workflows (Zheng et al., 2025; Lu et al., 2026). The working length of LLMs

has progressively expanded to 32K, 128K, and even to 1M tokens (Yang et al., 2025). However, full self-attention in vanilla transformers exhibits quadratic computational complexity with respect to context length, rendering both training and inference prohibitive for such extended contexts.

To address quadratic complexity, existing studies (Sun et al., 2025) have investigated *sparse attention* mechanisms, which selectively restrict the number of tokens each query attends to, reducing the complexity to sub-quadratic. The goal is to achieve full-attention-level performance with the efficiency of sparse attention. However, two issues remain unsolved:

**1) Attention Gap.** Training-free methods directly apply sparse attention to models trained with full attention (**Full-Sparse**)[1] (Xiao et al., 2024a;b; Jiang et al., 2024), exploiting inherent sparsity in attention. A metric that can quantify this approximation is **attention sparsity**: the proportion of total attention mass captured by selected tokens. We prove that the approximation error scales linearly with the dropped attention mass (Theorem 4.5), suggesting that higher attention sparsity yields closer approximation to full attention. However, the approximation error induced by the *mismatch between training and inference distributions* often incurs performance degradation (Yuan et al., 2025).

**2) Capability Gap.** Trainable sparse attention methods train and deploy with sparse attention (**Sparse-Sparse**) (Yuan et al., 2025; Lu et al., 2025), eliminating the distribution mismatch. However, as full attention is not involved during training as a reference, the performance gap between sparse and full attention is never explicitly defined and cannot be bounded. Whether Sparse-Sparse models can match Full-Full performance relies entirely on an implicit assumption that both could achieve the same output behavior after training on the same data. However, this assumption is brittle: Sparse-Sparse models can only attend to a subset of preceding tokens, weakening both forward and backward signals compared to full attention and resulting in lower performance than Full-Full models (Lu et al., 2025).

To address these issues, we observe that training with sparse or full attention alone is insufficient. We propose **SSA**

---

[*]Equal contribution   [1]Department of Informatics, King's College London, London, United Kingdom [2]Youtu Lab, Tencent, Shanghai, China (†work done during internship). Correspondence to: Yulan He <yulan.he@kcl.ac.uk>, Di Yin <endymecyyin@tencent.com>, Xing Sun <winfred-sun@tencent.com>.

*Proceedings of the $43^{rd}$ International Conference on Machine Learning*, Seoul, South Korea. PMLR 306, 2026. Copyright 2026 by the author(s).

---

[1]We use **(X-Y)** to denote attention configurations, where X and Y specify the attention mechanism used during training and inference, respectively.

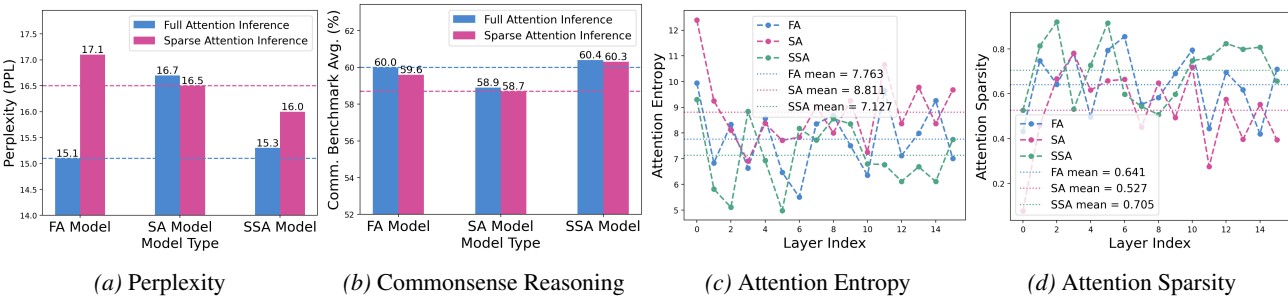

*Figure 1.* Performance and attention characteristics of 1B models. Under sparse inference, Sparse Attention Model (SA) achieves lower perplexity than Full Attention Model (FA) due to distribution mismatch, yet FA outperforms SA on benchmarks due to weaker training signals in SA. SSA resolves both issues, achieving the highest attention sparsity and best performance under both inference modes.

(Sparse Sparse Attention), a training framework that integrates both sparse and full attention while explicitly encouraging sparser attention distributions through attention-output alignment (Figure 2). At each training step, SSA randomly selects one of two attention modes with equal probability: **Sparse Attention** addresses the *Attention Gap* by directly optimizing for sparse inference, eliminating the train-inference distribution mismatch. **Full Attention** addresses the *Capability Gap* by providing complete gradient flow to all tokens, serving as a reference that enables attention sparsity to explicitly bound the approximation error. Furthermore, SSA computes the counterpart attention at each layer to perform bidirectional alignment, encouraging attention sparsity and tightening the bound: full-attention outputs are encouraged to match sparse-attention outputs, promoting inherently sparser distributions; sparse-attention outputs are regularized toward full-attention outputs, limiting drift from full-attention behavior.

Through this design, SSA attains substantially higher attention sparsity than both Full-Full and Sparse-Sparse baselines, translating into strong performance under both inference modes (Figure 1). Our contributions are as follows:

- We establish a theoretical framework showing that pure full training induces an attention gap while pure sparse training induces a capability gap, and prove that the approximation error between sparse and full attention scales linearly with the dropped attention mass.
- We propose **SSA**, a unified training framework that integrates both sparse and full attention with bidirectional alignment, addressing both gaps while promoting inherently sparser attention distributions.
- Extensive experiments demonstrate SSA achieves state-of-the-art performance across multiple benchmarks, supports flexible inference under varying sparsity budgets, and delivers superior long-context performance.

## 2. Related Work

**Training-Free Sparse Attention.** Training-free sparse attention leverages the intrinsic sparsity in the attention distribution, allowing it to closely approximate full attention while significantly reducing computational cost. A widely used form is sliding-window attention, which restricts each token to attend only to its local neighborhood (Child et al., 2019; Beltagy et al., 2020; Brown et al., 2020). StreamingLLM (Xiao et al., 2024b) further observes an "attention sink" phenomenon, where substantial attention mass is placed on the initial tokens, and incorporates these early tokens into the sparse attention set. Beyond fixed patterns, dynamic sparse attention selects informative segments based on the relevance between the query and attended keys. A common method is block-sparse attention (Xiao et al., 2024a; Jiang et al., 2024; Xu et al., 2025; Tang et al., 2024; Zhang et al., 2025), which partitions the context into blocks, then selects the top-$k$ important blocks for computation. Owing to the mismatch between training and inference, this type of methods often suffer from error compounding, especially on long context tasks (Hu et al., 2026). In a nutshell, we categorize all training-free sparse attention methods as Full-Sparse type.

**Trainable Sparse Attention.** To further improve sparse attention, MoBA (Lu et al., 2025) applies block-sparse attention during training, matching full-attention performance while being substantially more efficient. Through a gating module, NSA (Yuan et al., 2025) combines three complementary patterns, include coarse global attention, block-sparse attention, and sliding-window attention. Although NSA reports gains over full-attention models, subsequent analysis attributes these gains to the added gated-attention mechanism rather than the sparse design itself (Qiu et al., 2025); its multi-component structure also makes it difficult to adjust sparsity levels or revert to full attention at inference time. Concurrent with our work, InfLLM-v2 (Zhao et al., 2025) refines block selection via a two-level hierarchical mechanism, while DashAttention (Huang et al., 2026) replaces top-$k$ selection with an $\alpha$-entmax router (Peters et al., 2019; Gonçalves et al., 2025), yielding a query-adaptive number of selected blocks and a fully differentiable path from coarse routing to fine token-level attention. A separate line of work treats full attention as a teacher: Seer-

Attention (Gao et al., 2025; 2026a) and the concurrent DSA (DeepSeek-AI, 2025) distill full attention into sparse attention, and HySparse (Gao et al., 2026b) adopts a hybrid structure that directly reuses both the blocks and the KV caches of preceding full-attention layers. We refer to these natively trained approaches as the Sparse-Sparse paradigm.

In contrast to both the Full-Sparse and Sparse-Sparse paradigms, our method unifies sparse and full attention within a single training framework while explicitly encouraging higher attention sparsity. Although the aforementioned hybrid architectures and full-attention-distillation methods also involve full attention during training, they treat it as a fixed teacher; SSA instead aligns the two directions bidirectionally and explicitly drives full attention to become sparser. This design lets the model leverage the strengths of both paradigms during training and perform well under either inference mode.

## 3. Preliminary

**Full Attention.** In a standard transformer with softmax attention, each token attends to all preceding tokens:

$$h^{\text{full}}(t) = \text{softmax}\big[q(t)K^\top(:t)\big]V^\top(:t). \quad (1)$$

Here, $t$ denotes the position of the query token, $q(t)$ is its query vector, $K$ and $V$ are the key and value matrices, $(\cdot)^\top$ denotes matrix transposition. The notation $(:t)$ indexes all positions up to and including $t$. Since all tokens in the context are used to predict the next token, we refer to this formulation as **Full Attention**. This operation has quadratic complexity with respect to context length, thus becoming prohibitive for a large number of context tokens.

**Sparse Attention.** To address this computational bottleneck, sparse attention restricts each query to attend only to a subset of preceding keys and values. In this work, we focus on *block-sparse attention*, where tokens are partitioned into contiguous blocks, and only tokens within selected blocks are attended to. Following MoBA (Lu et al., 2025), we compute the block-level keys by mean-pooling the token keys within each block. For each query, we compute dot-product similarities with all preceding block keys and select the top-$k$ most relevant blocks. The tokens from selected blocks are concatenated to form reduced key and value matrices, denoted as $\tilde{K}$ and $\tilde{V}$, yielding sparse attention:

$$\tilde{K} = \{\, K(i) \mid i \in \text{Top-}k \text{ attention blocks}\,\},$$
$$h^{\text{sparse}}(t) = \text{softmax}\big[q(t)\tilde{K}^T(:t)\big]\tilde{V}^T(:t). \quad (2)$$

A key characteristic of sparse attention is that attention only relies on partial token set. Given a sequence of length $N$ partitioned into $n = N/s$ blocks of $s$ tokens each, selecting $k$ blocks yields a **receptive field** of $ks$ tokens with a **sparsity ratio** of $k/n$. The resulting computational complexity

becomes $\mathcal{O}(N \cdot ks) = \mathcal{O}(\frac{k}{n} \cdot N^2)$, reducing the quadratic cost of full attention proportionally to the sparsity ratio.

Despite its favorable sub-quadratic scaling in long-context settings, sparse attention remains a performance bottleneck, both when used purely at inference time and when trained natively, due to the degradation in modeling quality observed in prior work (Lu et al., 2025; Yuan et al., 2025).

## 4. Theoretical Framework of Sparse Attention

We develop a theoretical framework explaining why neither pure sparse nor pure full attention training suffices, and derive conditions under which sparse attention closely approximates full attention. Our analysis reveals two insights: (1) exclusive sparse training causes gradient deficiencies while exclusive full training induces distribution mismatch; and (2) the approximation error scales linearly with dropped attention mass, motivating inherently sparser distributions.

### 4.1. Problem Setup

The goal of sparse attention is to approximate full attention performance while using fewer tokens for efficiency. Let $h_{\text{Tr}}^{\text{Inf}}(t)$ denote the attention hidden representation for query token $t$, where $\text{Tr} \in \{\text{FA}, \text{SA}\}$ indicates training with full or sparse attention, and $\text{Inf} \in \{\text{full}, \text{sparse}\}$ indicates the inference mechanism. Ideally, a well-trained sparse attention model should achieve:

$$\sum_{t=1}^{T} \|h_{\text{FA}}^{\text{full}}(t) - h_{\text{SA}}^{\text{sparse}}(t)\| < \epsilon, \quad (3)$$

for some small $\epsilon > 0$, where $T$ denotes the sequence length.

### 4.2. Theoretical Framework

**Capability Gap: Insufficiency of Sparse-Only Training.** We identify two learning deficiencies induced by training exclusively with sparse attention.

**Proposition 4.1** (Learning Deficiency). *Training exclusively with sparse attention induces:*

1. ***Gradient Update Deficiency:** Full attention (Equation 1) computes $h^{full}(t)$ over all preceding keys $K(:t)$, so every key receives gradient. Sparse attention (Equation 2) computes $h^{sparse}(t)$ only over the selected subset $\tilde{K}(:t)$, so any dropped key receives zero gradient and is not learned at this step. Each query is thus shaped by only its selected keys, reducing the per-step learning signal versus full attention.*

2. ***Attention Suppression Deficiency:** Full attention normalizes over all preceding keys $K(:t)$, so raising one key's weight necessarily lowers every other key's—competitive pressure that lets the model push*

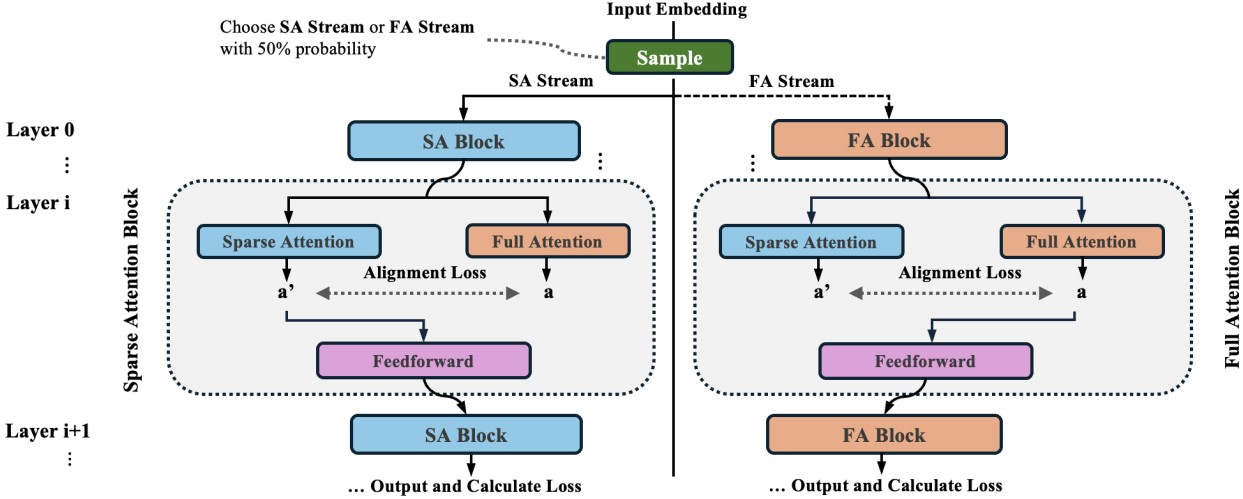

*Figure 2.* Illustration of the SSA training framework. At each iteration, the model has an equal probability of following either the Sparse Attention (SA) stream or the Full Attention (FA) stream. In the SA stream, the model learns sparse attention while aligning its output with a full-attention counterpart computed on the fly. Conversely, in the FA stream, the model learns full attention constrained by alignment with the corresponding sparse-attention output. For clarity, skip connections, normalization, and dropout layers are omitted in the figure.

*irrelevant keys toward zero. Sparse attention normalizes only over the selected subset $\tilde{K}(:t)$, removing dropped keys from this competition entirely. SA models thus have weaker pressure to globally suppress irrelevant tokens, a deficiency exposed when they are given the full key set instead of the selected subset.*

These deficiencies predict that sparse-trained models may: (1) underperform full-attention-trained models due to incomplete learning, compounding the information loss already incurred by sparse inference; and (2) fail to benefit from full attention at inference (an extrapolation setting) due to high-entropy attention patterns.

**Attention Gap: Insufficiency of Full-Only Training.** We formalize the distribution mismatch when training and inference use different attention mechanisms.

**Proposition 4.2** (Attention Mechanism Mismatch)**.** *Let $p_\theta^{\mathrm{full}}(t)$ and $p_\theta^{\mathrm{sparse}}(t)$ denote the next-token distributions obtained by running the same model with full and sparse attention at inference, respectively. Then the cross-entropy of the full-attention distribution relative to the sparse-attention distribution decomposes as:* [1]

$$H(p_\theta^{\mathrm{full}}(t), p_\theta^{\mathrm{sparse}}(t)) = H(p_\theta^{\mathrm{full}}(t)) + D_{\mathrm{KL}}(p_\theta^{\mathrm{full}}(t) \,\|\, p_\theta^{\mathrm{sparse}}(t)). \quad (4)$$

*Since $D_{KL} \geq 0$, switching a full-attention-trained model to sparse inference moves its output distribution away from full-attention behavior by $D_{\mathrm{KL}}(p_\theta^{\mathrm{full}}(t) \,\|\, p_\theta^{\mathrm{sparse}}(t))$, which vanishes only when the two distributions coincide.*

*Remark* 4.3. Switching a full-attention-trained model to sparse inference shifts its output distribution by a KL term.

A sparse-attention-trained model avoids this shift, since its training and inference attention coincide. Consequently, despite a full-attention-trained model performing stronger under full inference, a sufficiently large mismatch can make it *worse* than a sparse-trained model under sparse inference.

Propositions 4.1 and 4.2 collectively indicate that neither pure sparse nor pure full attention training suffices for high sparse-inference performance.

**Quantifying the approximation error.** To characterize whether sparse attention well-approximates full attention, we introduce a metric measuring the attention mass retained by sparse attention selection.

**Definition 4.4** (Attention Sparsity)**.** For query token $t$ with attention weights $\{a(t,j)\}_{j<t}$, the *attention sparsity* is:

$$\mathrm{AttnSparsity}(t) = \textstyle\sum_{j<t} a(t,j) \cdot \mathbf{1}[\mathrm{blk}(j) \in \mathcal{T}_k(t)], \quad (5)$$

where $\mathrm{blk}(j)$ denotes the block index of token $j$, $\mathcal{T}_k(t)$ is the set of top-$k$ blocks ranked by block-level attention scores, and $\mathbf{1}[\cdot]$ is the indicator function.

Higher attention sparsity indicates that sparse attention selection captures more of the total attention mass, suggesting closer approximation to full attention. We now bound the approximation error.

**Theorem 4.5** (Sparse Attention Error Bound)**.** *Let $\mathcal{S}(t)$ denote the selected tokens and $\mathcal{S}^c(t)$ the dropped tokens. Define the dropped attention mass as $\delta(t) = \sum_{j \in \mathcal{S}^c(t)} a_j^{\mathrm{full}}(t) = 1 - \mathrm{AttnSparsity}(t)$. Then:* [2]

$$\|h^{\mathrm{full}}(t) - h^{\mathrm{sparse}}(t)\| \leq \delta(t) \left(\max_{j \in \mathcal{S}^c(t)} \|v(j)\| + \|h^{\mathrm{sparse}}(t)\|\right),$$

---

[1]Proofs are provided in Appendix A.

[2]Proofs are provided in Appendix B.

*where $v(j)$ denotes the value vector of token $j$.*

This bound scales linearly with $\delta(t)$, motivating the learning of inherently sparser attention distributions to minimize approximation error under inference.

## 5. Sparse Sparse Attention (SSA)

**From theory to method.** Our theoretical analysis motivates decomposing the target gap (Equation 3) as:

$$\|h_{\text{FA}}^{\text{full}}(t) - h_{\text{SSA}}^{\text{sparse}}(t)\| \leq \underbrace{\|h_{\text{FA}}^{\text{full}}(t) - h_{\text{SSA}}^{\text{full}}(t)\|}_{\text{(I) Capability Gap}}$$
$$+ \underbrace{\|h_{\text{SSA}}^{\text{full}}(t) - h_{\text{SSA}}^{\text{sparse}}(t)\|}_{\text{(II) Attention Gap}}. \quad (6)$$

This motivates three design principles: (1) incorporate full attention training to minimize Term (I) (Proposition 4.1); (2) learn inherently sparser attention distributions to reduce Term (II) via the bound in Theorem 4.5; and (3) include sparse attention training to directly optimize for sparse inference (Proposition 4.2), improving performance beyond what this bound guarantees. Guided by these principles, we propose **SSA (Sparse Sparse Attention)** (Figure 2), a training framework that leverages the complementary strengths of sparse and full attention training while explicitly promoting attention sparsity, enabling sparse inference to match the capability of full-attention-trained models.

**Dual-stream training.** SSA alternates between full and sparse attention streams with equal probability. The full stream provides complete gradient flow to all tokens, addressing the learning deficiencies in Proposition 4.1 and minimizing Term (I). The sparse stream directly optimizes for sparse inference, which does not have the KL-divergence penalty in Proposition 4.2. By alternating rather than jointly computing both streams, we reduce computational cost by half while ensuring the model processes an equal number of tokens as the baselines for a fair comparison.

**Counterpart attention alignment.** To further reduce the *Attention Gap* (Term II), we enforce bidirectional alignment between the hidden representations produced by full and sparse attention. At each layer, we compute an auxiliary attention output using the counterpart attention mode, used only for alignment and not propagated to subsequent layers.

The alignment objective consists of two complementary components. A **sparsity loss** encourages full-attention outputs to mimic sparse-attention outputs:

$$\mathcal{L}_{\text{sparsity}} = \|h_{\text{full}} - \text{sg}(h_{\text{sparse}})\|, \quad (7)$$

where $h_{\text{full}}$ and $h_{\text{sparse}}$ are the hidden representations produced by full and sparse attention at a given layer, and $\text{sg}(\cdot)$

denotes the stop-gradient operator. This loss mirrors the left-hand side of Theorem 4.5, with an added stop-gradient on $h_{\text{sparse}}$ that drives $h_{\text{full}}$ toward $h_{\text{sparse}}$. We let full attention adapt to the sparse output rather than the reverse, since $h_{\text{sparse}}$ is constrained by top-$k$ selection and cannot recover the information it discards. This direction encourages full attention to concentrate on the selected blocks, which follows a sparser distribution that drops less weight under top-$k$ selection. Accordingly, we expect to reduce $\delta(t)$ and tighten the bound on Term (II). Provided $h_{\text{full}}$ remains non-degenerate, this shrinking gap pulls $h_{\text{sparse}}$ closer to $h_{\text{full}}$, improving the sparse attention performance.

A **commitment loss** regularizes sparse-attention outputs to remain close to full-attention outputs:

$$\mathcal{L}_{\text{commit}} = \|h_{\text{sparse}} - \text{sg}(h_{\text{full}})\|, \quad (8)$$

This is analogous to the commitment loss in VQ-VAE (van den Oord et al., 2017) and KL regularization in RLHF (Ouyang et al., 2022), anchoring the sparse attention to the representational space of full attention so that it does not drift too far from the full-attention behavior.

The total alignment loss combines both:

$$\mathcal{L}_{\text{align}} = \mathcal{L}_{\text{sparsity}} + \mathcal{L}_{\text{commit}}. \quad (9)$$

This bidirectional alignment encourages full attention to become inherently sparser (reducing $\delta(t)$) while maintaining the performance. Notably, value-level alignment is substantially more efficient than directly aligning attention distributions (Gao et al., 2025), which would require materializing dense attention maps incompatible with FlashAttention-2 (Dao, 2024). In practice, we use SmoothL1 loss for both components (Girshick, 2015; Shen et al., 2025).

**Overall objective.** The final training objective combines cross-entropy loss with alignment:

$$\mathcal{L} = \mathbb{E}_{\text{mode}\sim\{\text{full,sparse}\}}[\mathcal{L}_{\text{ce}}^{\text{mode}}] + \alpha\mathcal{L}_{\text{align}}, \quad (10)$$

where $\mathcal{L}_{\text{ce}}^{\text{mode}}$ is the cross-entropy loss under the sampled attention mode, and $\alpha$ is a weighting coefficient. The full algorithm is detailed in Algorithm 1 in the appendix.

**Efficiency analysis.** During inference, SSA performs identical sparse attention operations to MoBA (Lu et al., 2025), achieving $2\times$ speedup over full attention at 128k context length (Table A3). During training, although auxiliary attention is computed at each layer, it is not propagated to subsequent layers, increasing cost only marginally ($\sim 17\%$ when pretrained on 8k-context data, Table A2). More details about training and inference efficiency are in Appendix D.

## 6. Experimental Setup

**Pretraining Setup.** We follow the architecture and configuration of Llama-3.2-1B (Grattafiori et al., 2024) with two

*Table 1.* Comparison of different attention training methods on commonsense reasoning and perplexity under both full and sparse attention inference. The receptive field denotes the maximum number of accessible tokens during sparse-attention inference. SSA consistently outperforms all other methods on the benchmarks on average across all levels of sparsity. All results are averaged over 5 runs.

| Method | PIQA/% | HellaSwag/% | ARC-Easy/% | ARC-Challenge/% | Average/% ↑ | Wikitext PPL ↓ |
|---|---|---|---|---|---|---|
| **Full Attention Inference** | | | | | | |
| FullAttn | 74.17 ±0.37 | 58.22 ±0.14 | 69.29 ±0.55 | 38.35 ±0.38 | 60.01 ±0.19 | **15.13** |
| MoBA | 72.75 ±0.15 | 56.18 ±0.19 | 69.24 ±0.28 | 37.27±0.68 | 58.86 ±0.19 | 16.68 |
| **SSA** | **74.54** ±0.48 | **58.45** ±0.19 | **69.92** ±0.32 | **38.65** ±0.55 | **60.39** ±0.20 | 15.28 |
| **Sparse Attention Inference (Receptive Field = 256)** | | | | | | |
| FullAttn | 74.23 ±0.39 | 57.84 ±0.20 | 68.84 ±0.20 | 37.56 ±0.55 | 59.62 ±0.31 | 17.05 |
| MoBA | 72.64 ±0.34 | 56.39 ±0.21 | 68.83 ±0.41 | 37.05 ±0.30 | 58.73 ±0.16 | 16.54 |
| NSA | 73.23 ±0.30 | 58.15 ±0.32 | **69.80** ±0.34 | 36.71 ±0.58 | 59.47 ±0.20 | 16.02 |
| **SSA** | **74.53** ±0.55 | **58.40** ±0.08 | 69.62 ±0.30 | **38.53** ±0.64 | **60.27** ±0.22 | **15.96** |
| **Sparse Attention Inference (Receptive Field = 1024)** | | | | | | |
| FullAttn | 74.35 ±0.24 | 58.13 ±0.11 | 69.34 ±0.34 | 38.34 ±0.55 | 60.04 ±0.17 | 15.67 |
| MoBA | 72.99 ±0.26 | 56.26 ±0.15 | 69.16 ±0.37 | 36.14 ±0.33 | 58.64 ±0.14 | 16.02 |
| NSA | 73.57 ±0.29 | 58.39 ±0.08 | **70.07** ±0.21 | **38.69** ±0.49 | 60.18 ±0.15 | **15.48** |
| **SSA** | **74.53** ±0.10 | **58.50** ±0.11 | 69.97 ±0.29 | 38.57 ±0.51 | **60.39** ±0.15 | 15.51 |

*(a)* Commonsense Reasoning     *(b)* Perplexity     *(c)* LongBench     *(d)* NIAH@8k

*Figure 3.* Performance versus receptive-field size. SSA and FullAttn extrapolate well as more tokens become visible, whereas MoBA exhibits poor extrapolation. We scale the receptive field by fixing block size and varying top-k. Detailed scores are in Appendix I.

minor modifications (details in Appendix C). The model is pretrained on the SmolLM corpus (Allal et al., 2025) for 100B tokens with 8k context length, using a learning rate of 1e-3 with cosine decay and a global batch size of 3.15M tokens. To further assess long-context capability, we continue training for an additional 10B tokens at 32k context length with a learning rate of 1e-5 and a RoPE scaling factor of 4, using data sampled from DCLM (Li et al., 2024). The global batch size remains the same.

**Baselines.** We compare SSA against the following baselines: (1) **FullAttn**: the standard full attention mechanism. (2) **MoBA** (Lu et al., 2025): a trainable sparse attention method conceptually aligned with SSA's Sparse Attention Stream. (3) **NSA** (Yuan et al., 2025): a sparse attention framework with three components (compression, selection, and sliding window), aggregating them to form the final attention representation, where the selection module is analogous to MoBA. NSA has a larger effective receptive field than others: its compression module accesses the compressed representations of all tokens, and the sliding window attends

to a fixed number of local tokens. For each baseline, we train two configurations: one with a receptive field of 1024 (block size 32, top-32 blocks, and one with a receptive field of 256 (block size 16, top-16 blocks). For SSA, we report the 1024-receptive-field results obtained by extrapolating the 256-receptive-field model, as that model consistently performs better, this is likely due to inducing higher sparsity regularisation (see Section 7.5 for details). For NSA, we set the window size as 128 for both configurations.

**Evaluation.** We evaluate our models along two major dimensions. First, we assess performance on classical commonsense reasoning benchmarks: **PIQA** (Bisk et al., 2019), **Hellaswag** (Zellers et al., 2019), **ARC-Easy** (Clark et al., 2018), and **ARC-Challenge** (Clark et al., 2018). On each benchmark, we run 5 different seeds and report the average. We additionally measure word perplexity on **Wiki-Text** (Merity et al., 2017) with the context length capped at 8k. Second, we evaluate models on longer context tasks. Specifically, we use **LongBench** (Bai et al., 2024) (16 English benchmarks) to measure long-context understanding,

**Needle-in-A-Haystack** from *RULER* (Hsieh et al., 2024) to assess retrieval, and **PG19** (Rae et al., 2020) to compute long-context perplexity, where PPL is obtained via sliding-window evaluation with a stride of 256 (Press et al., 2021; 2022). For models not continually trained for 32k data, we employ a RoPE scaling factor of 4 for length extrapolation. All benchmarks are run using `lm-evaluate-harness` (Gao et al., 2024), and normalized accuracy is reported when applicable. All models are evaluated under both sparse and full attention modes at inference except NSA (discussed in Appendix F), enabling us to measure how well they generalize when given full KV-cache access. Unless otherwise specified, we use their receptive-field-256 variants for extrapolation. More details are shown in Appendix E.

## 7. Experimental Analysis

### 7.1. Language Modeling

As shown in Table 1 (last column), SSA achieves the lowest perplexity under sparser attention (RF=256), though it slightly lags behind NSA at RF=1024 due to NSA's sliding window module (see Appendix G). These results validate our theoretical framework. First, **supporting Proposition 4.2**, we observe that MoBA achieves lower perplexity than FullAttn under sparse-inference, despite FullAttn's superior full-inference performance. This confirms that pure full-attention training incurs a substantial distribution mismatch penalty when switching to sparse inference. Second, **supporting Theorem 4.5**, SSA exhibits a much smaller perplexity gap between sparse and full attention compared to FullAttn. As shown in Figure 1, SSA learns inherently sparser attention distributions, reducing the dropped attention mass $\delta(t)$ and thereby tightening the attention gap (Term II in Equation 6). Third, SSA's full-attention perplexity remains only marginally higher than FullAttn, indicating a small capability gap (Term I). Together, minimizing both terms yields SSA's superior sparse-inference performance.

### 7.2. Commonsense Reasoning

As shown in Table 1 (left five columns), SSA in general consistently outperforms all baselines under the same sparsity budget, and notably surpasses FullAttn while using a receptive field of only 256. Under sparse attention, MoBA underperforms FullAttn, **validating Proposition 4.1** about the learning deficiency, while SSA successfully overcomes this limitation by incorporating full attention into training. Interestingly, SSA achieves better downstream benchmark performance than FullAttn under full attention, despite having higher perplexity in the same setting. Our ablation results (Table 3) explain this phenomenon: removing the alignment loss (`NoAlignmentLoss`) substantially degrades benchmark performance, while training with full attention only but retaining the alignment loss (`FullRatio=1`) preserves the gains. This suggests that the sparser attention

distributions benefit downstream reasoning. A plausible explanation is that sparser attention encourages the model to focus on informative tokens while ignoring irrelevant ones. We note that most of these benchmarks involve contexts shorter than 1024 tokens, so the `RF=1024` setting effectively reduces to near `Full-Attention`. The practically relevant comparison is therefore at `RF=256`, where SSA outperforms all baselines by a clear margin.

### 7.3. Extrapolation between Different Levels of Sparsity

SSA extrapolates effectively across sparsity levels, exhibiting largely monotonic performance improvement across all four tasks as more tokens are included in sparse-attention computation (Figure 3). In contrast, MoBA fails to extrapolate on Perplexity and LongBench, and degrades at a receptive field of 4096 for Commonsense and NIAH. We attribute this to MoBA's insufficiently sparse attention distribution: when additional tokens become available, irrelevant tokens are not adequately suppressed, introducing noise rather than useful context. FullAttn exhibits similar extrapolation to SSA but performs worse at nearly all sparsity levels, indicating that SSA's higher attention sparsity bounds the gap between sparse and full attention more tightly.

### 7.4. Long Context Evaluation

We evaluate models' long context capabilities on three dimensions: perplexity, retrieval, and understanding; and evaluate models both extrapolated and continual-trained to 32k.

**Language Modeling.** On longer context tasks, Figure 4 shows that SSA consistently outperforms baselines under sparse attention on PG-19, consistent with the results in Table 1. Under full attention without continued training, SSA surprisingly outperforms FullAttn (Figure 4a), reversing the trend from Table 1. We attribute this to SSA's sparser attention distribution (Table A15): sparse attention patterns are more robust to RoPE scaling degradation, as evidenced by the smaller gap between RoPE-scaled and 32k-trained curves under sparse inference modes. Intuitively, attending to fewer and more local tokens reduces sensitivity to positional distortions introduced by RoPE scaling (see Appendix K for detailed analysis). After continued training, FullAttn recovers and surpasses SSA under full attention, though SSA retains its advantage under sparse inference. We exclude NSA from this comparison because its additional sliding-window module benefits perplexity specifically, conflating its sparse attention ability not shared by other methods. Under the windowless setting, NSA underperforms SSA on long-context perplexity (Table A7, Appendix G).

**Needle-in-a-Haystack (NIAH).** In Table 2, SSA is the strongest sparse attention method within 8k context length across all receptive fields and achieves 100% accuracy under full attention inference. This strong performance stems

*Table 2.* Evaluation in longer context lengths. Left: length extrapolation to 32K using RoPE scaling on models pretrained up to 8K tokens. Right: models continually trained to 32K tokens. SSA outperforms all baselines on LongBench and NIAH under all modes, except for NIAH under full-attention inference with continual training. Best results are **bolded**; full LongBench results are provided in Appendix J.

| Inference Mode | Receptive Field | Method | Length Extrapolation (RoPE Scaling) | | | | | | Continual Trained to 32k | | | | | |
|---|---|---|---|---|---|---|---|---|---|---|---|---|---|---|
| | | | Needle in A Haystack/% | | | | | LongBench/% | Needle in A Haystack/% | | | | | LongBench/% |
| | | | 4k | 8k | 16k | 32k | Avg | 32k | 4k | 8k | 16k | 32k | Avg | 32k |
| Full Attention | Full | FullAttn | **100** | **100** | 48 | **35.4** | 70.9 | 17.71 | **100** | **100** | 99.2 | **93.2** | **98.1** | 18.97 |
| | | MoBA | **100** | 32.4 | 2.2 | 0 | 33.7 | 13.62 | 99 | 73.2 | 5.4 | 1.2 | 44.7 | 14.47 |
| | | SSA | **100** | **100** | 83 | **35.4** | 79.6 | **18.32** | **100** | **100** | 99.8 | 70.6 | 92.6 | **19.55** |
| Sparse Attention | 256 | FullAttn | 48 | 9.2 | 4.2 | **2.8** | 16.1 | 14.75 | 53 | 11.8 | 5.4 | 3.6 | 18.5 | 14.96 |
| | | MoBA | 68.8 | 28.2 | **14.4** | 0 | 27.9 | 16.88 | 82 | 27.6 | **12.4** | 3 | 31.3 | 17.17 |
| | | NSA | 37.4 | 13.8 | 3.6 | 2.4 | 14.3 | 17.25 | 61 | 34.2 | 5.4 | **4.8** | 26.4 | 17.45 |
| | | SSA | **81.6** | **33.6** | 5.8 | 2.2 | **30.8** | **17.89** | **89.2** | **34.8** | 7.8 | 3.2 | **33.8** | **18.25** |
| Sparse Attention | 1024 | FullAttn | 58.2 | 21.4 | 7.2 | 4.4 | 22.8 | 16.72 | 59 | 25 | 9.2 | 5.2 | 24.6 | 16.84 |
| | | MoBA | 58.4 | 27.8 | **16** | **8** | 27.6 | 18.23 | 93.8 | 36.2 | **17.6** | **9.4** | 39.3 | 18.68 |
| | | NSA | 64.6 | 25 | 11.4 | 4 | 26.3 | 17.36 | 74.4 | 27.8 | 10.6 | 6.6 | 29.9 | 18.08 |
| | | SSA | **77.4** | **47.8** | 10 | 3.8 | **34.8** | **18.58** | **94** | **55.6** | 17.2 | 5.4 | **43.1** | **19.05** |

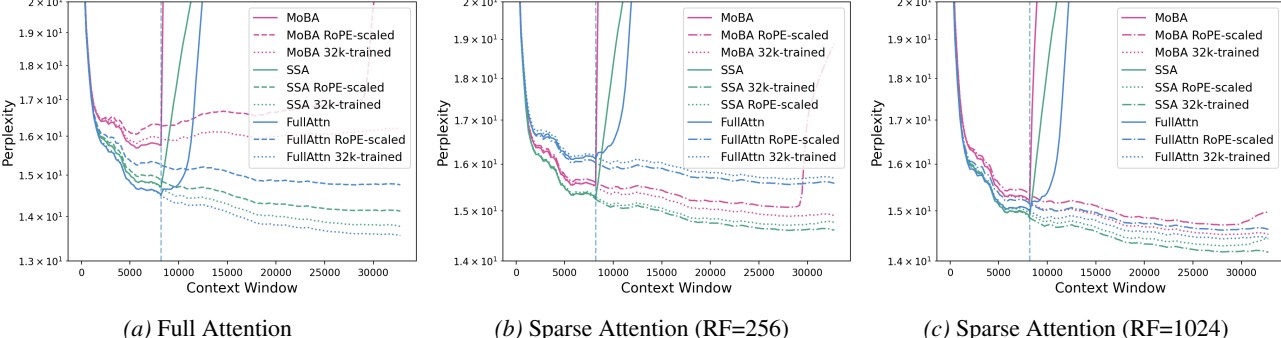

*(a)* Full Attention  *(b)* Sparse Attention (RF=256)  *(c)* Sparse Attention (RF=1024)

*Figure 4.* Perplexity across context lengths under different inference modes tested on PG-19. SSA achieves the lowest perplexity under sparse attention inference and exhibits the strongest robustness to RoPE scaling under full attention.

from SSA's alignment training: within 8k, full attention provides global awareness that guides sparse attention's feature learning and ranking, enabling reliable retrieval from arbitrary positions. Beyond 8k, SSA's performance decreases as this regime lies outside the alignment training distribution. In contrast, MoBA shows stronger retrieval at longer contexts due to its purely content-based block selection, which is more position-agnostic and thus generalizes better to out-of-distribution distances. Nevertheless, SSA maintains superior full-attention NIAH, whereas MoBA cannot effectively leverage full attention at inference. On average, SSA achieves the best performance in 5 out of 6 settings, offering practitioners the flexibility to choose sparse mode for efficiency or full attention for retrieval-intensive tasks.

**LongBench.** While perplexity and NIAH offer useful long-context diagnostics, they do not fully capture real-world long-context understanding: perplexity measures average prediction quality, and NIAH tests retrieval of a single fact. LongBench provides a more comprehensive evaluation across diverse tasks, requiring reasoning over long documents. Results from Table 2 confirm that SSA achieves

the best performance across all inference modes and context extension settings on LongBench.

### 7.5. Ablation Studies

We ablate SSA along several dimensions. All experiments are conducted with 300M-parameter models (Table A1) trained on 50B tokens.

**Sparsity Levels.** We observe that training with a larger receptive field (e.g., 16x32 or 16x64) does not improve SSA performance (Table 3). We hypothesize that a smaller receptive field imposes stronger structural constraints, providing more effective regularization for learning sparse attention patterns. Conversely, shrinking the receptive field too aggressively (e.g., 8x16) also fails to yield further benefits. These results suggest that an optimal receptive-field "sweet spot" is needed for the best performance.

**Sampling Ratio to the Full Attention Stream.** We vary the mixing ratio between the FA and SA streams. Moderate inclusion of the SA stream (e.g., `FullRatio = 0.75`) provides near-optimal perplexity, while placing more weight

*Table 3.* Ablation studies. `trainA×B` denotes training with top-k *A* and block size *B*, while `infA×B` indicate the same thing. `FullRatio` indicates the sampling ratio of the Full Attention Stream in Figure 2. `Only Full→Sparse` applies alignment only from full attention to sparse attention, while `Only Sparse→Full` applies alignment only in the reverse direction.

| Method | PPL ↓ | Comm. Avg. ↑ |
|---|---|---|
| **Baseline (SSA)** | 24.20 | 49.69 |
| **MoBA** | 24.51 | 48.81 |
| **Sparsity Level** | | |
| Baseline(*inf8×16*) | 25.26 | 49.69 |
| train8×16 | 25.97 | 48.88 |
| Baseline(*inf16×32*) | 23.52 | 49.78 |
| train16×32 | 23.60 | 49.32 |
| Baseline(*inf16×64*) | 23.23 | 49.96 |
| train16×64 | 23.22 | 49.16 |
| **Sampling Ratio to the Full Attention Stream** | | |
| FullRatio=1 | 25.16 | 49.58 |
| FullRatio=0.75 | 24.27 | 49.66 |
| FullRatio=0.5 (Baseline) | 24.20 | 49.69 |
| FullRatio=0.25 | 24.32 | 49.25 |
| FullRatio=0 | 24.40 | 49.00 |
| Random Each Layer | NaN | 0.00 |
| **Alignment Loss** | | |
| Only Full→Sparse | NaN | 0.00 |
| Only Sparse→Full | NaN | 0.00 |
| No Alignment Loss | 24.48 | 48.38 |
| **Alignment Loss Coefficient** | | |
| $\alpha = 5$ | 24.16 | 48.71 |
| $\alpha = 10$ (Baseline) | 24.20 | 49.69 |
| $\alpha = 20$ | 24.31 | 49.45 |
| **Alignment Loss Type** | | |
| L2 Loss | 24.31 | 49.27 |

on the FA stream generally yields better downstream benchmark results. Eliminating either stream leads to noticeable performance degradation.

**Alignment Loss.** Without the alignment loss $L_{\text{alignment}}$ performance drops considerably. We hypothesize that Full Attention and Sparse Attention prefer different model weights for best performance, abruptly switching between them harms stability. Using only one direction of the alignment loss also results in unstable training. We speculate this is due to asymmetric over-distillation: `Full→Sparse` forces the full attention path to overfit sparse patterns, degrading full-attention capability, while `Sparse→Full` has the opposite issue. Bi-directional alignment is therefore necessary to stabilize training.

**Alignment Loss Coefficient.** We observe that different weightings $\alpha$ affect performance, as they control the relative strength of the two loss terms. Hyperparameter tuning is required to balance the objectives.

**Alignment Loss Type.** We also experimented with L2 loss. Rather than tuning $\alpha$, we simply downscaled it from 10 to 5 based on the initial scale of the alignment loss relative to the cross-entropy loss. With this minimal adjustment, L2 performs slightly below our default SSA configuration. However, even this variant outperforms MoBA, suggesting that the choice of distance metric is flexible and any reasonable alignment loss should work for SSA.

## 8. Conclusion

We identified two fundamental limitations in sparse attention: pure sparse training induces a *capability gap* due to gradient deficiency on excluded tokens, while pure full training induces an *attention gap* from train-inference distribution mismatch. To address both, we proposed SSA, a unified framework integrating sparse and full attention streams with bidirectional alignment. The alignment loss encourages inherently sparser attention distributions, minimizing the dropped attention mass and tightening the approximation error bound. Experiments demonstrate SSA achieves state-of-the-art performance across diverse benchmarks, with a single model supporting flexible inference under both sparse and full attention modes. In future work, we aim to explore how SSA can enable efficient post-training.

## Impact Statement

This work improves the efficiency and robustness of sparse attention mechanisms for long-context modeling. In particular, it proposed a feasible training framework, called Sparse Sparse Attention (SSA), that addresses both the capability gap and the training-inference distribution mismatch in existing sparse attention approaches. SSA enables LLMs to achieve strong performance while reducing computational and memory costs at inference time. These improvements can facilitate broader access to long-context language modeling, particularly in resource-limited settings. More generally, our proposed theoretical insights may inform the design of efficient attention architectures in the future.

## Acknowledgments

This work was supported in part by the UK Engineering and Physical Sciences Research Council through a Turing AI Fellowship (grant no. EP/V020579/1, EP/V020579/2) and the Prosperity Partnership scheme (grant no. UKRI566). ZS is supported by a PhD studentship provided by the Chinese Scholarship Council.

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

## A. Proof of Proposition 4.2

*Proof.* Fix a query position $t$ and its preceding context. Running the same model with full and sparse attention at inference yields two next-token distributions over the vocabulary, $p_\theta^{\text{full}}(t)$ and $p_\theta^{\text{sparse}}(t)$. Let $y$ denote a token drawn from $p_\theta^{\text{full}}(t)$, and let $\mathbb{E}_{\text{full}}[\cdot]$ denote the expectation over $y \sim p_\theta^{\text{full}}(t)$. The expectation is taken under $p_\theta^{\text{full}}(\cdot \mid t)$, the model's full-attention distribution, since we measure the drift of sparse inference relative to this reference; this is precisely the choice under which the cross-entropy gap equals a KL divergence. With this convention, the entropy of the full-attention distribution and its cross-entropy relative to the sparse-attention distribution are

$$H\big(p_\theta^{\text{full}}(t)\big) = \mathbb{E}_{\text{full}}\big[ -\log p_\theta^{\text{full}}(y)\big],$$
$$H\big(p_\theta^{\text{full}}(t),\, p_\theta^{\text{sparse}}(t)\big) = \mathbb{E}_{\text{full}}\big[ -\log p_\theta^{\text{sparse}}(y)\big].$$

Both expectations are taken over the *same* distribution $p_\theta^{\text{full}}(t)$, conditioned on the same context. By the definition of the KL divergence,

$$
\begin{aligned}
D_{\text{KL}}\big(p_\theta^{\text{full}}(t) \,\big\|\, p_\theta^{\text{sparse}}(t)\big) &= \mathbb{E}_{\text{full}}\left[\log \frac{p_\theta^{\text{full}}(y)}{p_\theta^{\text{sparse}}(y)}\right] \\
&= \mathbb{E}_{\text{full}}\big[ -\log p_\theta^{\text{sparse}}(y)\big] - \mathbb{E}_{\text{full}}\big[ -\log p_\theta^{\text{full}}(y)\big] \\
&= H\big(p_\theta^{\text{full}}(t),\, p_\theta^{\text{sparse}}(t)\big) - H\big(p_\theta^{\text{full}}(t)\big).
\end{aligned}
$$

Rearranging yields the claim:

$$H\big(p_\theta^{\text{full}}(t),\, p_\theta^{\text{sparse}}(t)\big) = H\big(p_\theta^{\text{full}}(t)\big) + D_{\text{KL}}\big(p_\theta^{\text{full}}(t) \,\big\|\, p_\theta^{\text{sparse}}(t)\big).$$

$\square$

## B. Proof of Theorem 4.5

*Proof.* We decompose the full attention output into contributions from selected tokens $\mathcal{S}(t)$ and dropped tokens $\mathcal{S}^c(t)$:

$$h^{\text{full}}(t) = \sum_{j \in \mathcal{S}(t)} a^{\text{full}}(t, j) v(j) + \sum_{j \in \mathcal{S}^c(t)} a^{\text{full}}(t, j) v(j). \tag{11}$$

For any token $j \in \mathcal{S}(t)$, the full and sparse attention weights are related by:

$$a^{\text{full}}(t, j) = \frac{\exp(s(t, j))}{\sum_{k<t} \exp(s(t, k))}, \quad a^{\text{sparse}}(t, j) = \frac{\exp(s(t, j))}{\sum_{k \in \mathcal{S}(t)} \exp(s(t, k))},$$

where $s(t, j)$ denotes the unnormalized attention score from query position $t$ to key position $j$. Their ratio yields:

$$\frac{a^{\text{full}}(t, j)}{a^{\text{sparse}}(t, j)} = \frac{\sum_{k \in \mathcal{S}(t)} \exp(s(t, k))}{\sum_{k<t} \exp(s(t, k))} = 1 - \delta(t),$$

where $\delta(t) = \sum_{j \in \mathcal{S}^c(t)} a^{\text{full}}(t, j)$ is the dropped attention mass. Thus, for any $j \in \mathcal{S}(t)$:

$$a^{\text{full}}(t, j) = (1 - \delta(t))\, a^{\text{sparse}}(t, j). \tag{12}$$

Substituting Eq. (12) into the first term of Eq. (11):

$$\sum_{j \in \mathcal{S}(t)} a^{\text{full}}(t, j) v(j) = (1 - \delta(t)) \sum_{j \in \mathcal{S}(t)} a^{\text{sparse}}(t, j) v(j) = (1 - \delta(t))\, h^{\text{sparse}}(t).$$

Substituting back into Eq. (11) and rearranging:

$$h^{\text{full}}(t) - h^{\text{sparse}}(t) = \sum_{j \in \mathcal{S}^c(t)} a^{\text{full}}(t, j) v(j) - \delta(t)\, h^{\text{sparse}}(t).$$

Taking norms and applying the triangle inequality:

$$\|h^{\text{full}}(t) - h^{\text{sparse}}(t)\| \leq \left\| \sum_{j \in \mathcal{S}^c(t)} a^{\text{full}}(t, j) v(j) \right\| + \delta(t) \|h^{\text{sparse}}(t)\|. \tag{13}$$

For the first term, applying the triangle inequality and using the non-negativity of attention weights:

$$\left\| \sum_{j \in \mathcal{S}^c(t)} a^{\text{full}}(t, j) v(j) \right\| \leq \sum_{j \in \mathcal{S}^c(t)} a^{\text{full}}(t, j) \|v(j)\| \leq \delta(t) \max_{j \in \mathcal{S}^c(t)} \|v(j)\|.$$

Substituting into Eq. (13) and factoring out $\delta(t)$ yields the result:

$$\|h^{\text{full}}(t) - h^{\text{sparse}}(t)\| \leq \delta(t) \left( \max_{j \in \mathcal{S}^c(t)} \|v(j)\| + \|h^{\text{sparse}}(t)\| \right).$$

$\square$

---

**Algorithm 1** SSA Dual-Stream Training with A Bi-directional Alignment

---

**Input:** embeddings $x_0$, targets $y$, number of layers $L$, alignment weight $\alpha$, routing prob. $p_{\text{FA}}$
**if** training **then**
    goFA $\leftarrow$ Bernoulli($p_{\text{FA}}$)
**else**
    goFA $\leftarrow$ False
**end if**
MainAttn $\leftarrow$ FullAttn **if** goFA **else** SparseAttn
AuxAttn $\leftarrow$ SparseAttn **if** goFA **else** FullAttn
$x \leftarrow x_0;$    $\mathcal{L}_{\text{align}} \leftarrow 0$
**for** $l = 1$ **to** $L$ **do**
    $h \leftarrow \text{Norm1}_l(x)$
    $q \leftarrow Q_l(h);$  $k \leftarrow K_l(h);$  $v \leftarrow V_l(h)$
    $h_{\text{main}} \leftarrow \text{MainAttn}(q, k, v)$
    $h_{\text{aux}} \leftarrow \text{AuxAttn}(q, k, v)$
    $\mathcal{L}_{\text{sparse}} \leftarrow \|h_{\text{main}} - \text{sg}(h_{\text{aux}})\|$
    $\mathcal{L}_{\text{commit}} \leftarrow \|h_{\text{aux}} - \text{sg}(h_{\text{main}})\|$
    $\mathcal{L}_{\text{align}} \leftarrow \mathcal{L}_{\text{align}} + \mathcal{L}_{\text{sparse}} + \mathcal{L}_{\text{commit}}$
    $x \leftarrow x + O_l(\text{Gate}_l(h) \odot h_{\text{main}})$
    $h \leftarrow \text{Norm2}_l(x)$
    $x \leftarrow x + \text{FFN}_l(h)$
**end for**
$\mathcal{L}_{\text{align}} \leftarrow \mathcal{L}_{\text{align}}/L;$    $x \leftarrow \text{LayerNorm}(x)$
logits $\leftarrow \text{LMHead}(x)$
**if** training **then**
    $\mathcal{L}_{\text{ce}} \leftarrow \text{CrossEntropy}(\text{logits}, y)$
    **return** logits, $\mathcal{L}_{\text{ce}} + \alpha \cdot \mathcal{L}_{\text{align}}$
**else**
    **return** logits
**end if**

---

## C. Implementation Details

The model configurations are summarized in Table A1. We adopt the open-source NSA implementation[3] and use the Liger Kernel (Hsu et al., 2025) for acceleration.

---

[3] https://github.com/fla-org/native-sparse-attention

*Table A1.* Model configurations for the 1B and 300M parameter models.

| Config Field | 1B Model | 300M Model |
|---|---|---|
| Block Size | 16 | 16 |
| Block Counts | 16 | 16 |
| Hidden Size | 2048 | 1024 |
| Intermediate Size | 8192 | 4096 |
| Num Hidden Layers | 16 | 16 |
| Num Attention Heads | 32 | 16 |
| Num KV Heads | 2 | 1 |
| Head Dim | 64 | 64 |
| Max Position Embeddings | 131072 | 131072 |
| Vocabulary Size | 128256 | 128256 |
| BOS Token | 128000 | 128000 |
| EOS Token | 128001 | 128001 |
| RMSNorm Eps | 1e-5 | 1e-5 |
| Hidden Activation | SiLU | SiLU |
| Attention Bias | false | false |
| MLP Bias | false | false |
| Attention Dropout | 0.0 | 0.0 |
| Initializer Range | 0.02 | 0.02 |
| Pretraining TP | 1 | 1 |
| Tie Word Embeddings | true | true |
| Torch Dtype | bfloat16 | bfloat16 |
| RoPE Base $\theta$ | 500,000 | 500,000 |
| *RoPE Scaling (Long Context Evaluation Only)* | | |
| Scaling Factor | 4.0 | 4.0 |
| Low / High Freq Factor | 1.0 / 4.0 | 1.0 / 4.0 |
| Scaling Type | llama3 | llama3 |
| Original Max Position | 8192 | 8192 |

We annotate the two minor modifications: (1) We reduce the number of key-value heads to 2 to adapt block-sparse attention implementation and accelerate training; (2) We adopt Gated Attention (Qiu et al., 2025) to mitigate the attention sink-this mechanism is also implicitly employed in NSA (Yuan et al., 2025)-so we apply it to all methods for fair comparison.

## D. Efficiency Analysis

We report training and inference time statistics in Table A2 and Table A3, respectively.

For training, FullAttn achieves the shortest time due to its highly optimized implementation. SSA incurs only 17% more time than MoBA, indicating that the dynamic attention computation in each layer introduces minimal overhead. Interestingly, the pre-training time for RF=256 is shorter than that for RF=1024, but this trend reverses during continual training. This can be explained by the two-stage computation in sparse attention: the first stage computes block-level attention scores without softmax, and the second stage computes attention over the selected tokens. Denoting the number of tokens involved in each stage as $N$ and $M$, we can approximate the computational complexity as $N^2 + M^2$. As shown in Table A4, this analysis explains the observed efficiency patterns.

At inference, SSA matches MoBA's speed under sparse mode and equals FullAttn when full attention is triggered. While the sparse mode offers a modest gain at 8k, the advantage widens as context grows—delivering 2× speed-up at 128k length.

*Table A2.* Wall clock time (GPU-Hours) comparison for training different attention mechanisms. SSA incurs only slightly higher training time ($< 18\%$) than MoBA.

| Method | Receptive Field | Pre-training (8k) | Continual (32k) |
|---|---|---|---|
| FullAttn | - | 792 | 112 |
| MoBA | 256 | 856 | 136 |
|  | 1024 | 928 | 128 |
| NSA | 256 | 976 | 160 |
|  | 1024 | 1040 | 144 |
| SSA | 256 | 1008 | 176 |
|  | 1024 | 1088 | 168 |

*Table A3.* Wall clock time (ms) comparison for different attention mechanisms during inference. Sparse attention achieves noticeable speedup at longer context lengths, despite less optimized implementations. Measured for batch size of 1 averaged over 30 runs.

| Method | RF Config | 8k | 16k | 32k | 64k | 128k |
|---|---|---|---|---|---|---|
| FullAttn | — | 204.9 | 356.3 | 798.3 | 2178.7 | 7029.2 |
| Sparse Attention | RF=256 | 198.1 | 327.2 | 674.9 | 1723.9 | 5138.0 |
|  | RF=1024 | 199.5 | 317.0 | 595.2 | 1321.8 | 3411.9 |
| NSA | RF=256 | 202.6 | 336.8 | 695.3 | 1771.4 | 5204.1 |
|  | RF=1024 | 204.8 | 326.7 | 621.0 | 1365.4 | 3495.9 |

*Table A4.* Computational complexity of different sparse attention configurations under varying context lengths.

| Context Length | Receptive Field | #Block-level Tokens | #Selected Tokens | Approx. Complexity |
|---|---|---|---|---|
| 8192 | 256 | 512 | 256 | 327k |
| 8192 | 1024 | 256 | 1024 | 1.11M |
| 32768 | 256 | 2048 | 256 | 4.26M |
| 32768 | 1024 | 1024 | 1024 | 2.1M |

## E. Evaluation and Datasets

The statistics and n-shot samples used is shown in Table A5. The dataset statistics and few-shot configurations are summarized in Table A5. For reproducibility, we use fixed random seeds (1, 12, 123, 1234, and 12345) across the five runs on commonsense reasoning benchmarks. For LongBench and NIAH, we use greedy decoding and report results from a single run due to computational constraints—our sparse attention kernel does not currently support KV caching, resulting in slow decoding. Note that NIAH results may vary slightly across runs as the evaluation data is synthesized dynamically.

For perplexity evaluation, we use `lm-evaluation-harness` for Wikitext, reporting word-level perplexity as implemented by the framework. For PG-19 long-context perplexity, we implement sliding-window evaluation with a stride of 256 (Press et al., 2021; 2022). Due to the large number of samples, we evaluate on 1000 randomly selected samples using seed 42 and convert the loss to perplexity directly via the exponential relation.

*Table A5.* Evaluation Benchmark Statistics

| Evaluation Benchmark | N-Shot | Data Size |
|---|---|---|
| PIQA | 3 | 1,838 |
| Hellaswag | 10 | 10,042 |
| ARC-Easy | 25 | 2,376 |
| ARC-Challenge | 5 | 1,172 |
| LongBench | - | 4,750 |
| Needle-in-A-Haystack | - | 500 cases per test |

*Table A6.* NSA extrapolated to full attention results. `cmp`, `fa`, and `swa` denote compression, full attention, and sliding window, respectively. We do not evaluate the commonsense reasoning for those methods exploding the perplexity.

| Configuration | PIQA | HellaSwag | ARC-E | ARC-C | AVG | Wiki8K |
|---|---|---|---|---|---|---|
| cmp+fa+swa | 73.24 | 58.13 | 70.13 | 37.06 | 59.64 | 15.86 |
| cmp+fa+fa | – | – | – | – | – | 60.88 |
| fa+fa+swa | – | – | – | – | – | 72.05 |
| fa+fa+fa | – | – | – | – | – | 191.3 |

## F. NSA Is Not Suitable for Full Attention Evaluation

NSA aggregates outputs from three independent sparse attention modules, each capturing information at different levels: (1) **Compression**: maintains a compressed representation with a full view of all preceding tokens; (2) **Selection**: a top-$k$ block selection module that operates identically to MoBA; and (3) **Sliding Window**: a local attention module that attends to the most recent tokens. Excluding any module would impair performance due to distribution mismatch.

In principle, all three modules can be extrapolated to full attention. However, as shown in Table A6, replacing all three with full attention (`fa+fa+fa`) yields extremely high perplexity (191.3). To identify which modules fail to extrapolate, we test three additional configurations: `cmp+fa+swa`, `cmp+fa+fa`, and `fa+fa+swa` as we already know the selection module can be extrapolated to full attention well. The results show that only the selection module extrapolates well to full attention, and the compression and sliding window modules both fail. Since two out of three modules cannot extrapolate reliably, we exclude NSA from full attention evaluation.

## G. NSA with and without Sliding Window

We investigate whether NSA's strong perplexity performance stems from its sparse attention mechanism or its sliding window component, which is not used in other sparse attention methods. In our experiments, NSA is trained with a receptive field of 256 tokens, while the sliding window size is 128 tokens. As shown in Table A7, removing the sliding window leads to degradation across most downstream tasks, with notable drops in Hellaswag (-1.18) and ARC-E (-1.1). More critically, Table A8 reveals that without a sliding window, NSA suffers severe perplexity degradation at longer context lengths with RoPE scaling and continual-training, and SSA can consistently outperform NSA without the sliding window. **These results suggest that NSA's reported perplexity advantages largely depend on the sliding window module, which contributes additional local tokens, rather than its sparse attention design, making direct comparisons with other sparse attention methods that lack this component potentially misleading.**

*Table A7.* Comparison of NSA with and without the sliding window module on commonsense reasoning and perplexity.

| Method | PIQA | Hellaswag | ARC-E | ARC-C | Average | PPL ↓ |
|---|---|---|---|---|---|---|
| NSA-window | 73.23 ±0.3 | **58.15** ±0.32 | **69.8** ±0.34 | **36.71** ±0.58 | **59.47** ±0.2 | **16.02** |
| NSA-nowindow | **73.44** ±0.43 | 56.97 ±0.24 | 68.7 ±0.45 | 36.48 ±0.53 | 58.9 ±0.21 | 16.3 |

*Table A8.* Long-context perplexity after RoPE scaling on PG-19. The best perplexities are **bolded**, and the second best perplexities are underlined. SSA consistently outperforms NSA without a sliding window on long-context perplexity evaluation.

| Method | 4K | 8K | 16K | 32K |
|---|---|---|---|---|
| NSA-window | **15.66** | **15.33** | **15.20** | **15.00** |
| NSA-nowindow | 16.29 | 16.01 | 15.99 | 18.16 |
| SSA | 16.07 | 15.78 | 15.67 | 15.53 |
| NSA-window-32k-trained | **15.93** | **15.6** | **15.46** | **15.21** |
| NSA-nowindow-32k-trained | 16.18 | 15.87 | 15.73 | 15.51 |
| SSA-32k-trained | 16.01 | 15.7 | 15.56 | 15.32 |

## H. KL Divergence

We analyze the relationship between attention approximation quality and downstream performance by measuring two metrics: (1) KL divergence between sparse and full attention at the final layer, and (2) attention sparsity. Table A9 reports these metrics alongside perplexity and average commonsense reasoning scores for SSA, MoBA, and FullAttn. The results show that SSA achieves the smallest KL divergence, which we attribute to the alignment loss.

*Table A9.* Comparison of KL divergence, attention sparsity, perplexity, and benchmark accuracy for SSA, MoBA, and FullAttn under 1B-parameter settings.

| | | AttnSparsity | | PPL | | Commonsense Avg. /% | |
|---|---|---|---|---|---|---|---|
| **Method** | **KL Divergence** | **Sparse** | **Full** | **Sparse** | **Full** | **Sparse** | **Full** |
| SSA | 0.0661 | 0.677 | 0.705 | 15.97 | 15.28 | 60.27 | 60.39 |
| MoBA | 0.1024 | 0.546 | 0.527 | 16.54 | 16.68 | 58.73 | 58.86 |
| FullAttn | 0.1486 | 0.610 | 0.641 | 17.05 | 15.13 | 59.62 | 60.01 |

## I. Spasity Extrapolation Results

The detailed scores for commonsense reasoning and Longbench are reported in Table A10 and Table A11 respectively.

*Table A10.* Commonsense reasoning and perplexity under sparse inference across varying receptive fields. Both MoBA and SSA are trained with RF=256 but extrapolate to larger RFs. Results are averaged over 5 runs with different seeds.

| Method | PIQA/% | HellaSwag/% | ARC-Easy/% | ARC-Challenge/% | Average/% ↑ | Wikitext PPL ↓ |
|---|---|---|---|---|---|---|
| **Sparse Attention Inference (Receptive Field = 512)** | | | | | | |
| FullAttn | 74.30 ±0.40 | 58.12 ±0.16 | 68.96 ±0.53 | 38.34 ±0.48 | 59.93 | 16.09 |
| MoBA | 72.79 ±0.17 | 56.24 ±0.19 | 69.23 ±0.29 | 37.20 ±0.46 | 58.87 | 16.36 |
| **SSA** | **74.49** ±0.40 | **58.48** ±0.14 | **70.08** ±0.20 | **38.77** ±0.66 | **60.46** | **15.62** |
| **Sparse Attention Inference (Receptive Field = 1024)** | | | | | | |
| FullAttn | 74.30 ±0.40 | 58.12 ±0.15 | 69.39 ±0.54 | 38.34 ±0.37 | 60.04 | **15.53** |
| MoBA | 72.79 ±0.17 | 56.20 ±0.18 | 69.39 ±0.26 | 37.44 ±0.79 | 58.96 | 16.41 |
| **SSA** | **74.49** ±0.40 | **58.44** ±0.09 | **70.08** ±0.33 | **38.57** ±0.66 | **60.40** | 15.44 |
| **Sparse Attention Inference (Receptive Field = 2048)** | | | | | | |
| FullAttn | 74.30 ±0.40 | 58.12 ±0.16 | 69.39 ±0.54 | 38.28 ±0.34 | 60.02 | **15.23** |
| MoBA | 72.79 ±0.16 | 56.19 ±0.17 | 69.40 ±0.24 | 37.46 ±0.70 | 58.96 | 16.55 |
| **SSA** | **74.49** ±0.40 | **58.45** ±0.10 | **70.08** ±0.33 | **38.43** ±0.51 | **60.36** | 15.39 |
| **Sparse Attention Inference (Receptive Field = 4096)** | | | | | | |
| FullAttn | 74.30 ±0.40 | 58.12 ±0.16 | 69.39 ±0.54 | 38.28 ±0.34 | 60.02 | **15.15** |
| MoBA | 72.63 ±0.36 | 56.15 ±0.19 | 69.22 ±0.23 | 37.25 ±0.68 | 58.81 | 16.59 |
| **SSA** | **74.58** ±0.47 | **58.57** ±0.11 | **69.99** ±0.18 | **38.79** ±0.51 | **60.48** | 15.26 |

## J. Full Longbench Results

Only the average scores are reported in Table 2. The full results evaluated in Longbench is in Table A12 and Table A13.

*Table A11.* LongBench Evaluation under sparse inference across varying receptive fields. Both MoBA and SSA are trained with RF=256 but extrapolate to larger RFs.

| Category | Dataset | Receptive Field = 512 | | | Receptive Field = 1024 | | |
|---|---|---|---|---|---|---|---|
| | | FullAttn | MoBA | SSA | FullAttn | MoBA | SSA |
| Single Doc | NarQA | 2.09 | 2.76 | 3.91 | 1.96 | 2.85 | 2.92 |
| | Qasper | 6.76 | 7.74 | 7.12 | 6.85 | 8.07 | 7.27 |
| | MFQA | 14.97 | 13.12 | 15.28 | 16.72 | 12.51 | 16.64 |
| Multi Doc | HotpotQA | 5.50 | 7.06 | 6.94 | 5.80 | 6.93 | 7.10 |
| | 2WikiQA | 8.77 | 9.71 | 9.79 | 8.87 | 9.70 | 10.18 |
| | MuSiQue | 3.58 | 3.86 | 4.06 | 3.85 | 3.82 | 3.68 |
| Summary | GovReport | 8.17 | 11.55 | 9.48 | 10.15 | 12.23 | 9.52 |
| | QMSum | 18.28 | 16.83 | 17.38 | 18.60 | 15.58 | 17.38 |
| | MultiNews | 14.81 | 12.78 | 17.06 | 15.42 | 14.08 | 16.49 |
| Few-shot | TREC | 28.5 | 45.5 | 51.5 | 32 | 48 | 56.5 |
| | TriviaQA | 40.88 | 37.62 | 48.80 | 45.77 | 38.62 | 50.22 |
| | SAMSum | 33.39 | 23.62 | 33.78 | 34.82 | 19.34 | 34.28 |
| Synthetic | PsgCount | 3.09 | 3.25 | 1.29 | 3.11 | 3.09 | 1.03 |
| | PsgRe-en | 3.91 | 3.89 | 4.22 | 3.76 | 4.29 | 3.94 |
| Code | LCC | 30.44 | 38.97 | 29.70 | 32.14 | 37.23 | 29.41 |
| | RepoBen | 38.58 | 40.62 | 39.36 | 40.39 | 39.28 | 39.46 |
| Average ↑ | | 16.36 | 17.33 | **18.73** | 17.51 | 17.23 | **19.13** |

| Category | Dataset | Receptive Field = 2048 | | | Receptive Field = 4096 | | |
|---|---|---|---|---|---|---|---|
| | | FullAttn | MoBA | SSA | FullAttn | MoBA | SSA |
| Single Doc | NarQA | 1.96 | 2.50 | 4.60 | 2.77 | 1.30 | 5.06 |
| | Qasper | 7.13 | 7.05 | 7.59 | 7.24 | 5.88 | 6.72 |
| | MFQA | 16.77 | 12.86 | 15.45 | 16.69 | 12.04 | 16.41 |
| Multi Doc | HotpotQA | 6.05 | 7.01 | 7.13 | 6.89 | 6.56 | 7.54 |
| | 2WikiQA | 8.52 | 8.83 | 10.26 | 10.35 | 8.10 | 9.58 |
| | MuSiQue | 3.94 | 4.25 | 3.76 | 3.81 | 3.49 | 3.98 |
| Summary | GovReport | 10.77 | 12.29 | 10.09 | 11.81 | 11.93 | 9.35 |
| | QMSum | 17.85 | 14.12 | 17.76 | 16.99 | 9.73 | 18.11 |
| | MultiNews | 14.81 | 13.83 | 16.03 | 14.73 | 13.80 | 15.06 |
| Few-shot | TREC | 37.5 | 44.5 | 54 | 41 | 39 | 55.5 |
| | TriviaQA | 46.29 | 38.66 | 51.00 | 46.17 | 35.47 | 48.62 |
| | SAMSum | 33.45 | 13.50 | 34.47 | 34.14 | 9.86 | 33.76 |
| Synthetic | PsgCount | 3.09 | 2.84 | 1.19 | 3.28 | 2.41 | 1.30 |
| | PsgRe-en | 3.56 | 3.75 | 4.34 | 3.68 | 4.21 | 3.89 |
| Code | LCC | 31.98 | 37.51 | 29.29 | 32.03 | 37.10 | 29.05 |
| | RepoBen | 39.74 | 38.33 | 39.24 | 38.57 | 37.07 | 39.54 |
| Average ↑ | | 17.78 | 16.38 | **19.14** | 18.13 | 14.87 | **18.97** |

*Table A12.* LongBench evaluation with per-dataset scores for length extrapolated models with RoPE scaling.

| Category | Dataset | Full | | | Receptive Field = 256 | | | | Receptive Field = 1024 | | | |
|---|---|---|---|---|---|---|---|---|---|---|---|---|
| | | FA | MoBA | SSA | FA | MoBA | NSA | SSA | FA | MoBA | NSA | SSA |
| Single Doc | NarQA | 2.02 | 1.33 | 3.41 | 1.89 | 4.12 | 2.31 | 2.45 | 2.01 | 3.24 | 3.2 | 3.23 |
| | Qasper | 7.46 | 5.82 | 7.19 | 5.84 | 7.52 | 7.06 | 6.12 | 6.9 | 6.52 | 6.58 | 6.69 |
| | MFQA | 16.77 | 11.26 | 16.33 | 14.24 | 13.75 | 13.62 | 16.66 | 15.91 | 15.14 | 16.99 | 15.6 |
| Multi Doc | HotpotQA | 7.37 | 6.02 | 6.88 | 5.29 | 6.42 | 7.03 | 6.42 | 5.23 | 6.04 | 6.35 | 6.43 |
| | 2WikiQA | 10.18 | 7.36 | 10.24 | 8.99 | 9.64 | 9.5 | 10.03 | 8.78 | 8.74 | 8.9 | 10.71 |
| | MuSiQue | 3.97 | 3.62 | 3.69 | 3.46 | 3.49 | 4.86 | 3.57 | 2.98 | 3.43 | 4.11 | 3.34 |
| Summary | GovReport | 11.48 | 12.18 | 8.76 | 6.69 | 10.24 | 10.57 | 9.1 | 8.82 | 6.33 | 9.76 | 8.6 |
| | QMSum | 15.36 | 5.55 | 17.12 | 17.85 | 17.7 | 18.83 | 17.06 | 18.24 | 17.67 | 18.13 | 17.81 |
| | MultiNews | 14.66 | 14.04 | 15.13 | 12.84 | 13.05 | 12.39 | 15.26 | 15.13 | 11.01 | 8.58 | 16.41 |
| Few-shot | TREC | 41 | 34 | 49.5 | 19 | 40 | 39.5 | 48.5 | 28.5 | 47.5 | 34.5 | 52 |
| | TriviaQA | 44.18 | 29.39 | 48.85 | 35.88 | 37.55 | 37.54 | 44.72 | 41.83 | 46.64 | 46.88 | 48.75 |
| | SAMSum | 31.56 | 9.01 | 33.39 | 30.51 | 23.74 | 30.51 | 32.57 | 33.93 | 35.07 | 31 | 33.27 |
| Synthetic | PsgCount | 3.17 | 1.99 | 1.07 | 3.18 | 3.32 | 3.3 | 0.97 | 3.14 | 1.02 | 3.18 | 1.27 |
| | PsgRe-en | 3.53 | 3.11 | 4.03 | 3.74 | 2.75 | 3.4 | 3.42 | 4.07 | 4.58 | 3.38 | 4.01 |
| Code | LCC | 32.75 | 37.16 | 29.05 | 29.97 | 38.19 | 36.56 | 29.5 | 31.97 | 37.21 | 35.82 | 29.35 |
| | RepoBen | 37.93 | 36.1 | 38.52 | 36.67 | 38.52 | 38.98 | 39.83 | 40.11 | 41.51 | 40.45 | 39.75 |
| Average ↑ | | 17.71 | 13.62 | **18.32** | 14.75 | 16.88 | 17.25 | **17.89** | 16.72 | 18.23 | 17.36 | **18.58** |

*Table A13.* LongBench evaluation with per-dataset scores for continual-trained models.

| Category | Dataset | Full | | | Receptive Field = 256 | | | | Receptive Field = 1024 | | | |
|---|---|---|---|---|---|---|---|---|---|---|---|---|
| | | FA | MoBA | SSA | FA | MoBA | NSA | SSA | FA | MoBA | NSA | SSA |
| Single Doc | NarQA | 2.35 | 2.24 | 4.87 | 1.92 | 3.66 | 2.32 | 4.61 | 1.88 | 4.5 | 2.41 | 3.71 |
| | Qasper | 8.31 | 6.64 | 7.77 | 5.12 | 6.95 | 7.23 | 6.54 | 7.54 | 7.59 | 7.21 | 7.42 |
| | MFQA | 19.15 | 11.3 | 15.95 | 13.78 | 13.54 | 14.04 | 15.45 | 14.87 | 15.93 | 16.73 | 15.8 |
| Multi Doc | HotpotQA | 7 | 6.22 | 7.56 | 5.14 | 6.2 | 6.47 | 6.25 | 5.24 | 6.76 | 6.74 | 6.69 |
| | 2WikiQA | 8.14 | 7.71 | 11 | 9.18 | 9.65 | 8.5 | 9.83 | 9.39 | 9.23 | 9.17 | 10.03 |
| | MuSiQue | 3.63 | 3.44 | 3.77 | 3.22 | 3.8 | 4.04 | 3.82 | 3.46 | 3.64 | 3.89 | 3.38 |
| Summary | GovReport | 12.3 | 12.09 | 12.41 | 7 | 10.48 | 12.25 | 9.94 | 8.52 | 9.32 | 12.53 | 10.61 |
| | QMSum | 18.78 | 4.96 | 18.17 | 18.72 | 19.15 | 19.7 | 17.87 | 18.73 | 17.73 | 17.65 | 18.42 |
| | MultiNews | 15.72 | 14.03 | 20.43 | 12.95 | 12.8 | 13.59 | 16.87 | 16.51 | 12 | 12.3 | 19.95 |
| Few-shot | TREC | 49.5 | 40 | 52 | 25.5 | 42 | 41.5 | 48.5 | 31.5 | 48 | 38.5 | 50.5 |
| | TriviaQA | 46.98 | 32.96 | 50.45 | 35.08 | 37.23 | 38.67 | 46.24 | 40.63 | 45.65 | 50.01 | 49.67 |
| | SAMSum | 35.27 | 11.39 | 34.89 | 30.21 | 24.56 | 32.2 | 33.23 | 34.25 | 35.56 | 32.54 | 34.34 |
| Synthetic | PsgCount | 3.02 | 1.74 | 0.85 | 3.18 | 3.52 | 3.2 | 0.69 | 3.21 | 0.71 | 3.18 | 0.87 |
| | PsgRe-en | 3.96 | 3.08 | 3.8 | 3.16 | 4.1 | 3.48 | 3.95 | 5.06 | 5 | 3.92 | 3.4 |
| Code | LCC | 30.31 | 35.97 | 28.67 | 27.84 | 37.65 | 33.96 | 29.13 | 29.98 | 36.12 | 32.63 | 29.13 |
| | RepoBen | 39.14 | 37.68 | 40.21 | 37.43 | 39.43 | 38.12 | 39.12 | 38.65 | 41.14 | 39.84 | 40.9 |
| Average ↑ | | 18.97 | 14.47 | **19.55** | 14.96 | 17.17 | 17.45 | **18.25** | 16.84 | 18.68 | 18.08 | **19.05** |

# K. Sparse Attention is Robust to RoPE Scaling

RoPE scaling prevents perplexity explosion beyond the pre-trained context window (Figure 4a) by extending the periods of low-frequency bands, thereby eliminating aliasing (see Appendix K.1 for details). However, this scaling impairs overall perplexity by distorting the learned frequency representations. Interestingly, sparse attention is more robust to this degradation. Figure 4 shows that perplexity degradation follows: sparse 256 tokens < sparse 1024 tokens < full attention. We attribute this to two factors. First, sparse attention attends to fewer tokens overall, limiting cumulative exposure to distorted positional encodings. Second, the selected tokens are skewed toward local context (<8k positions) (Appendix K.2 and Figure A4), where RoPE scaling induces less distortion.

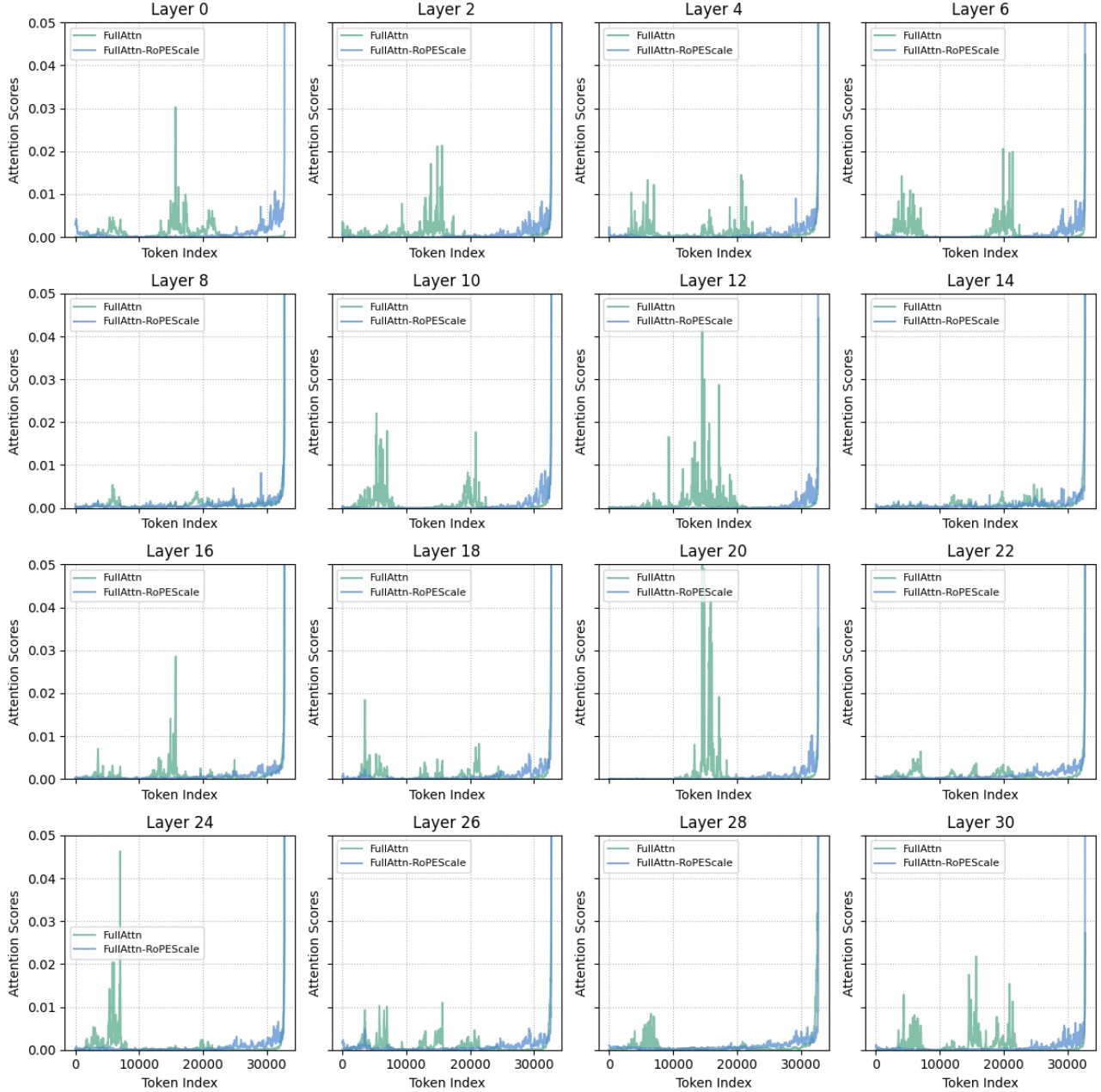

*Figure A1.* Attention score distributions for FullAttn and FullAttn with RoPE scaling at a context length of 32k.

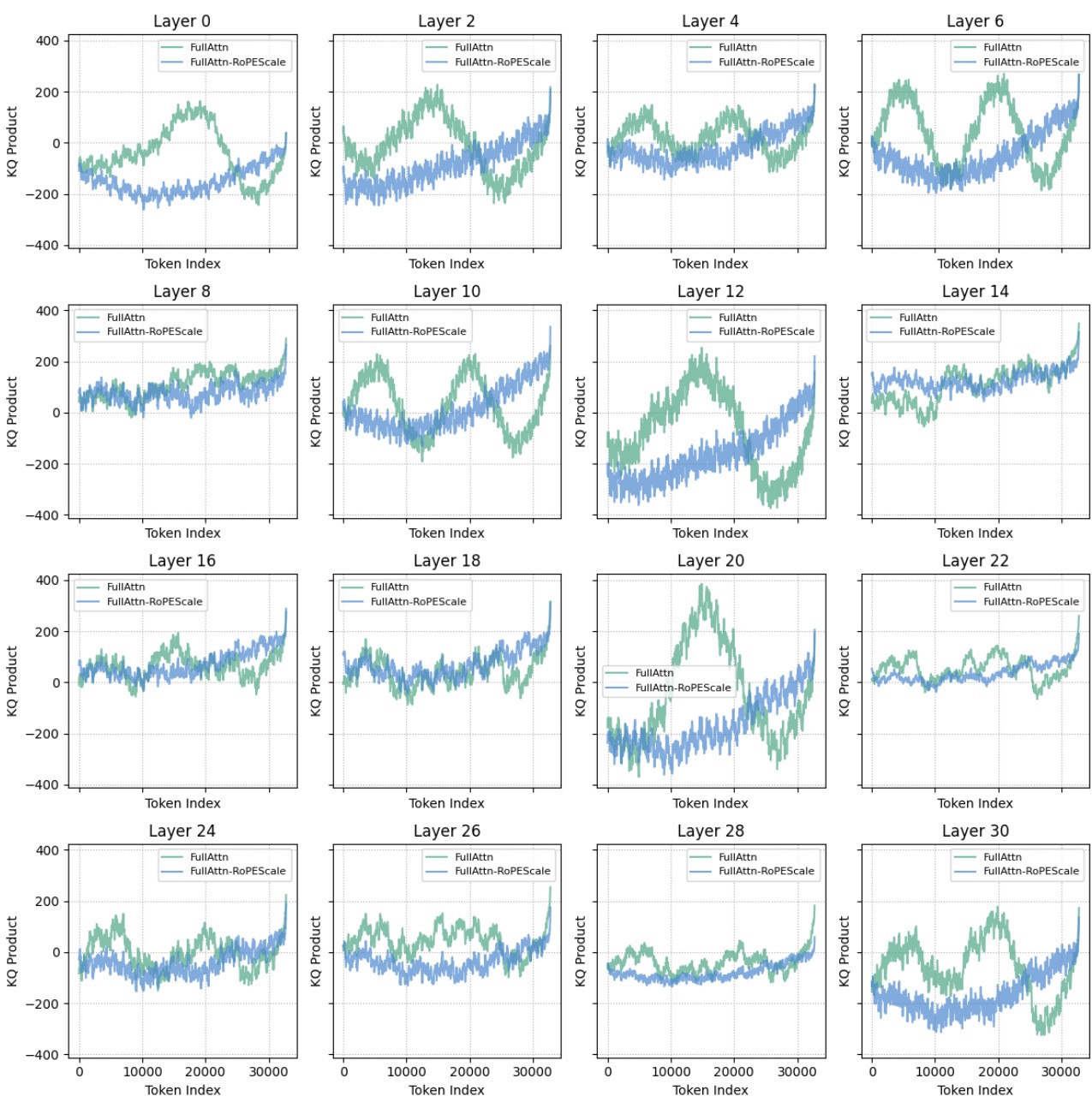

*Figure A2.* The KQ Product distributions for FullAttn and FullAttn with RoPE scaling at a context length of 32k. The KQ Product is the value before the softmax operation when calculating the attention.

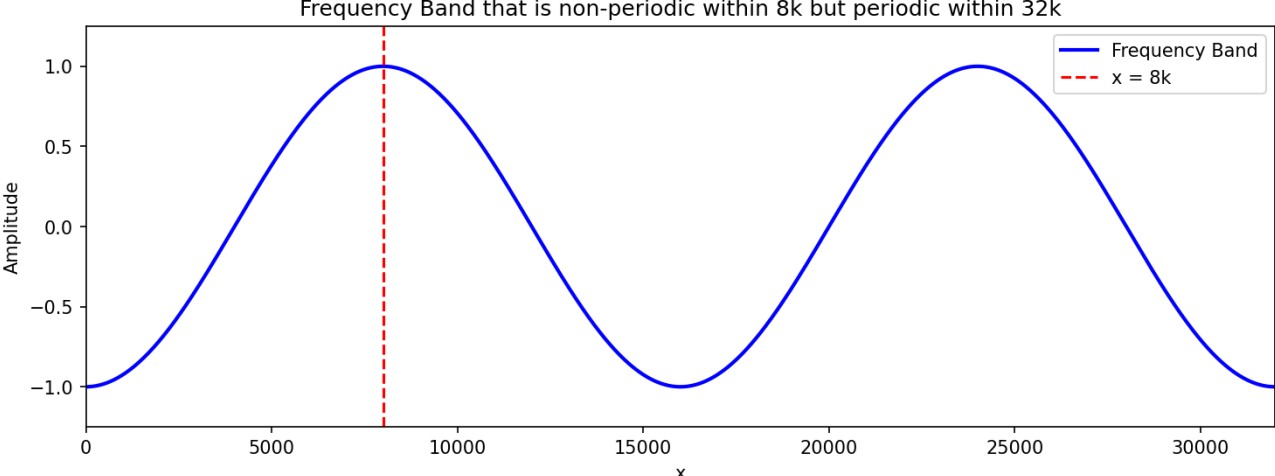

*Figure A3.* A Frequency Band that is non-periodic within the pre-training context length but periodic within a longer context length.

*Table A14.* RoPE frequency periods before and after scaling.

| Dimension | Before Scaling | After Scaling |
|:---:|:---:|:---:|
| 0 | $6.28 \times 10^0$ | $6.28 \times 10^0$ |
| 1 | $9.47 \times 10^0$ | $9.47 \times 10^0$ |
| 2 | $1.43 \times 10^1$ | $1.43 \times 10^1$ |
| 3 | $2.15 \times 10^1$ | $2.15 \times 10^1$ |
| 4 | $3.24 \times 10^1$ | $3.24 \times 10^1$ |
| 5 | $4.88 \times 10^1$ | $4.88 \times 10^1$ |
| 6 | $7.36 \times 10^1$ | $7.36 \times 10^1$ |
| 7 | $1.11 \times 10^2$ | $1.11 \times 10^2$ |
| 8 | $1.67 \times 10^2$ | $1.67 \times 10^2$ |
| 9 | $2.52 \times 10^2$ | $2.52 \times 10^2$ |
| 10 | $3.79 \times 10^2$ | $3.79 \times 10^2$ |
| 11 | $5.72 \times 10^2$ | $5.72 \times 10^2$ |
| 12 | $8.62 \times 10^2$ | $8.62 \times 10^2$ |
| 13 | $1.30 \times 10^3$ | $1.30 \times 10^3$ |
| 14 | $1.96 \times 10^3$ | $1.96 \times 10^3$ |
| *— Period = 2048: Below unscaled, above linearly interpolated —* | | |
| 15 | $2.95 \times 10^3$ | $4.24 \times 10^3$ |
| 16 | $4.44 \times 10^3$ | $9.64 \times 10^3$ |
| 17 | $6.70 \times 10^3$ | $2.19 \times 10^4$ |
| *— Period = 8192: Above scaled by factor $4\times$ —* | | |
| 18 | $1.01 \times 10^4$ | $4.04 \times 10^4$ |
| 19 | $1.52 \times 10^4$ | $6.08 \times 10^4$ |
| 20 | $2.29 \times 10^4$ | $9.16 \times 10^4$ |
| 21 | $3.45 \times 10^4$ | $1.38 \times 10^5$ |
| 22 | $5.20 \times 10^4$ | $2.08 \times 10^5$ |
| 23 | $7.84 \times 10^4$ | $3.14 \times 10^5$ |
| 24 | $1.18 \times 10^5$ | $4.73 \times 10^5$ |
| 25 | $1.78 \times 10^5$ | $7.12 \times 10^5$ |
| 26 | $2.68 \times 10^5$ | $1.07 \times 10^6$ |
| 27 | $4.04 \times 10^5$ | $1.62 \times 10^6$ |
| 28 | $6.09 \times 10^5$ | $2.44 \times 10^6$ |
| 29 | $9.18 \times 10^5$ | $3.67 \times 10^6$ |
| 30 | $1.38 \times 10^6$ | $5.53 \times 10^6$ |
| 31 | $2.08 \times 10^6$ | $8.34 \times 10^6$ |

### K.1. Aliased Frequency Bands in Length Extrapolation

When a pre-trained model is directly applied to a 32k context length without adaptation, its attention distribution exhibits anomalous patterns—sometimes manifesting as dual peaks in the middle of the context, as shown in Figure A1. This phenomenon is distinct from the well-known attention sink problem. Examining the QK product (i.e., the logits before softmax) in Figure A2 reveals a clear periodic wave pattern, which arises from frequency aliasing in RoPE's positional encoding. The root cause lies in the model's limited exposure during training. When trained on 8k contexts, the model never observes how certain frequency bands behave at longer distances. Specifically, frequency bands that appear non-periodic within 8k positions become periodic when extrapolated to 32k, as illustrated in Figure A3. This aliasing causes the model to perceive distant positions as "local," leading to the perplexity explosion beyond 8k shown in Figure 4. RoPE scaling resolves this issue by stretching the periods of low-frequency bands by a factor of 4 (Table A14). Consequently, frequency bands that were non-periodic within 8k remain non-periodic at 32k, eliminating the aliasing artifacts visible in Figures A1 and A2.

### K.2. Local Token Preference for Long Context

We compute the distribution of distances between selected keys and the current query token for FullAttn (Figure A4), MoBA (Figure A5), and SSA (Figure A6). All three methods exhibit a preference for local tokens, which are less affected by RoPE scaling in lower frequency bands. We also compute sparsity and entropy for FullAttn, MoBA, and SSA under four different context lengths with RoPE scaling, as shown in Table A15. SSA consistently achieves higher sparsity and lower entropy.

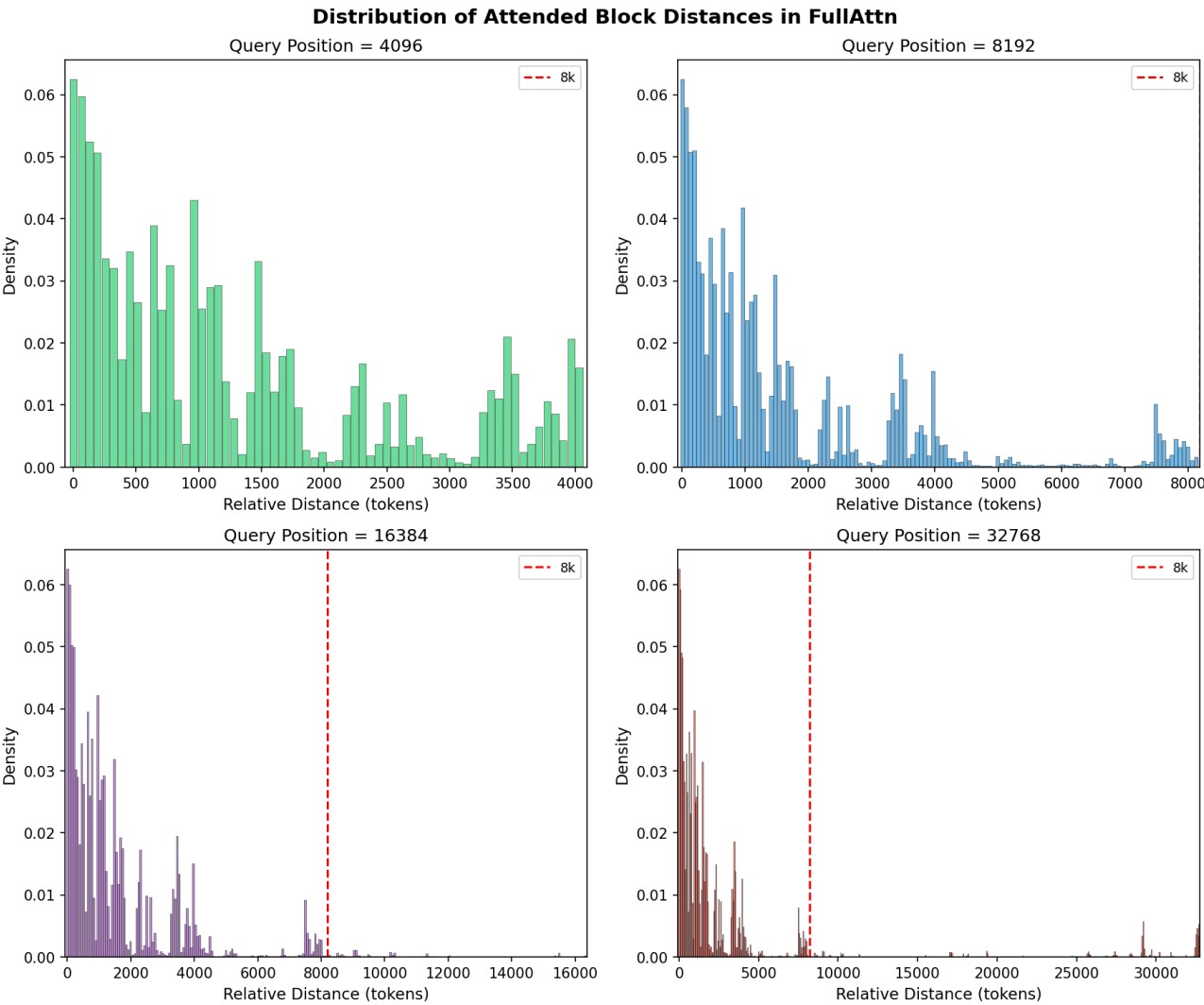

*Figure A4.* The histogram of the distance from the attended token to the query for FullAttn in sparse attention inference mode.

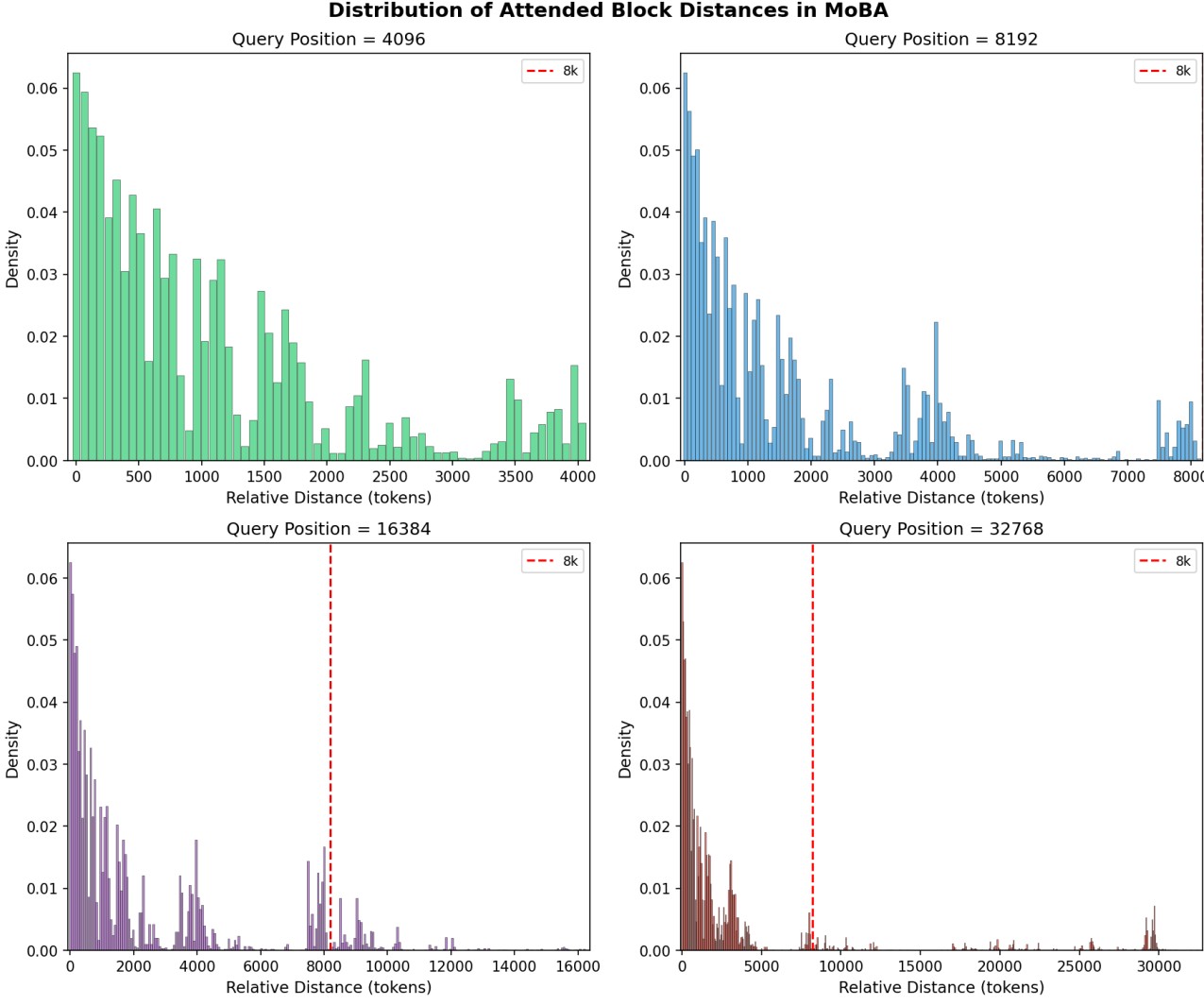

*Figure A5.* The histogram of the distance from the attended token to the query for MoBA in sparse attention inference mode.

*Table A15.* Attention patterns under full attention inference during long-context inference. SSA exhibits consistently sparser attention patterns.

| Context | Method | AttnSparsity ↑ | AttnEntropy ↓ |
|---------|--------|----------------|---------------|
| 4k | FullAttn | 0.612 | 8.89 |
| | MoBA | 0.606 | 9.01 |
| | SSA | **0.782** | **7.63** |
| 8k | FullAttn | 0.553 | 9.49 |
| | MoBA | 0.541 | 9.69 |
| | SSA | **0.737** | **8.07** |
| 16k | FullAttn | 0.543 | 9.70 |
| | MoBA | 0.490 | 10.28 |
| | SSA | **0.714** | **8.28** |
| 32k | FullAttn | 0.511 | 10.21 |
| | MoBA | 0.423 | 11.30 |
| | SSA | **0.707** | **8.51** |

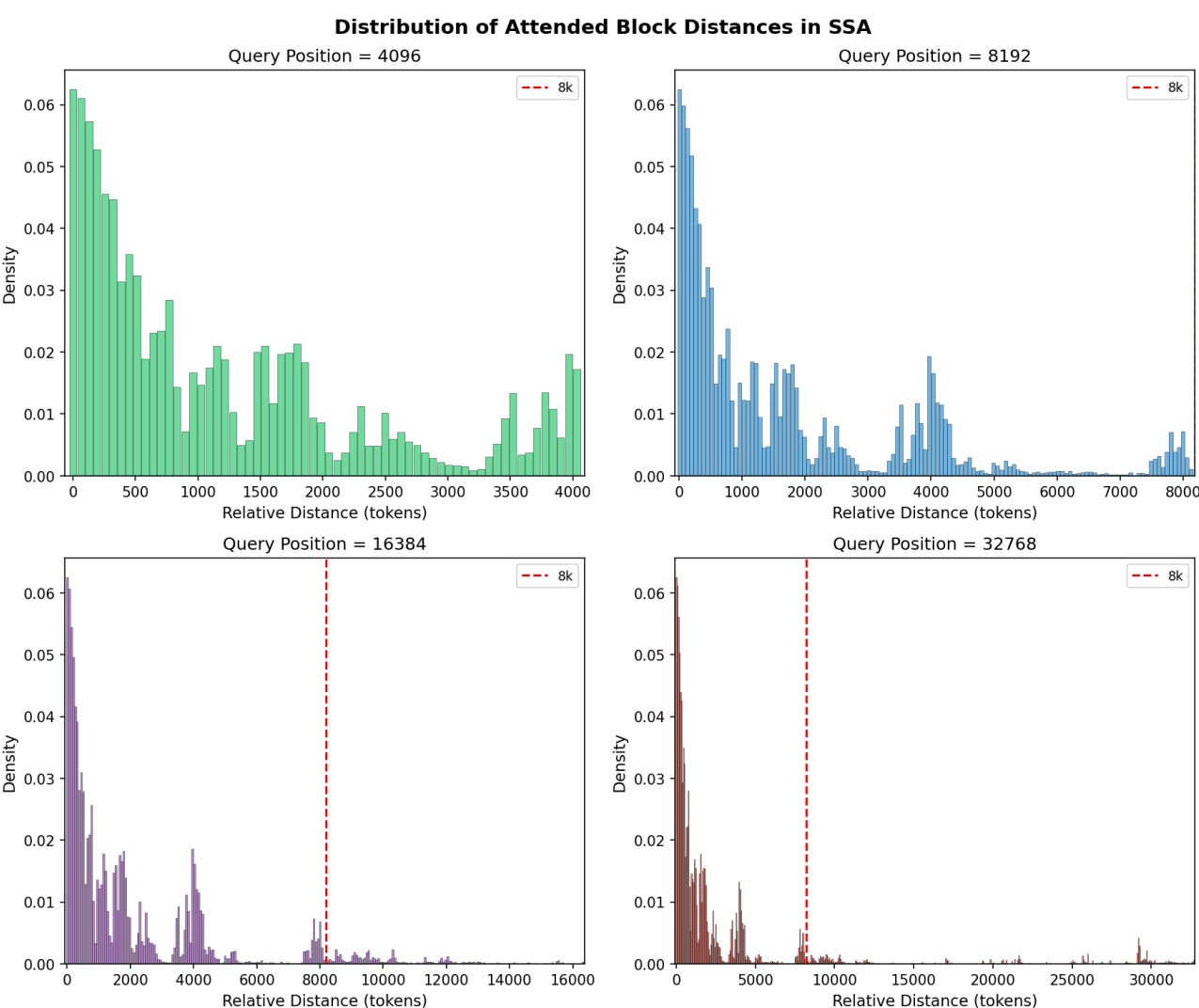

*Figure A6.* The histogram of the distance from the attended token to the query for SSA in sparse attention inference mode.

