# OpenReview forum: "SSA: Sparse Sparse Attention by Aligning Full and Sparse Attention Outputs in Feature Space"
_ICML.cc/2026/Conference — ICML 2026 regular_

### Official Review · Reviewer_hxyn · 2026-03-09

**Soundness:** 4
**Presentation:** 4
**Significance:** 3
**Originality:** 3
**Overall Recommendation:** 5
**Confidence:** 4

**Summary:**

The paper proposes SSA (Sparse Sparse Attention), a novel training framework designed to mitigate the quadratic complexity of self-attention in LLMs. The authors theoretically diagnose two primary issues in existing sparse attention paradigms: the "capability gap" (gradient deficiency in pure sparse training) and the "attention gap" (distribution mismatch in pure full-attention training). To address this, SSA implements a dual-stream training approach—randomly alternating between full and sparse attention at each step—coupled with a bidirectional feature alignment loss. The authors prove that the approximation error between sparse and full attention is strictly bounded by the dropped attention mass, which SSA's alignment effectively minimizes. Extensive experiments on a 1B parameter model trained on 100B tokens demonstrate that SSA establishes state-of-the-art performance across diverse benchmarks under both sparse and full inference modes, showcasing excellent context length extrapolation.

**Compliance With Llm Reviewing Policy:**

Affirmed.

**Final Justification:**

My concerns are addressed. I keep my positive score.

**Key Questions For Authors:**

1. While the <18% training overhead is acceptable at 8k/32k context lengths, does the memory/time overhead of the unpropagated auxiliary stream become a bottleneck when training on extreme context lengths?
2. Have you conducted any preliminary experiments at larger model scales (e.g., >7B parameters) to confirm that the bidirectional alignment loss does not cause instability or mode collapse during training?
3. Have you tried training on longer context lengths to validate the scalability of SSA?

**Limitations:**

Yes.

**Strengths And Weaknesses:**

Pros:

- The theoretical formalization of the "attention gap" and "capability gap" (Propositions 4.1 and 4.2) is insightful.
- The dual-stream training coupled with a feature-space alignment loss is intuitive, conceptually clean.
- The paper is well-written, featuring rich ablations and analyses that effectively demonstrate the efficacy of the proposed method.

Cons:

- The authors specifically utilize the Needle-in-A-Haystack (NIAH) evaluation from the RULER benchmark, but omit the rest of the dataset. The results would be more convincing if a full evaluation of RULER were included.
- The proposed dual-stream training pipeline requires two attention passes in a single training step, which could lead to substantial computational overhead. Although the authors claim in Appendix D that the time overhead is under 18% , this is based on a maximum training length of 32K. Scaling this to extreme context lengths, such as 128K or longer, remains a significant concern.
- The scalability of the method is largely unaddressed. The experiments are conducted on a 1B parameter model with a maximum training length of 32K. It is unclear whether the model's performance would collapse due to the incorporation of the auxiliary loss at larger scales, or if it remains performant at extended context lengths.

---

> ### Author Rebuttal · Authors · 2026-03-30
>
> We thank the reviewer for the detailed and constructive feedback.
>
> ### 1. Complete RULER Evaluation
> We ran a complete RULER evaluation (13 tasks in total) @32k (RF=256 extrapolated). SSA consistently outperforms baselines in both inference modes. We will add complete RULTER results in the final version.
> |Inference Mode|Full/%|Sparse/%|
> |-|-|-|
> |SSA|25.48|12.63|
> |MoBA|2.57|3.61|
> |FullAttn|17.71| 5.69|
> ### 2. Computational Cost w.r.t. Incorporating Full Attention in Long-Context Training
> *At 128k context length, SSA trains are only ~30% longer than MoBA with comparable memory consumption, which we consider tractable.* Moreover, SSA's framework is **flexible**: the two streams need not be strictly full vs. sparse attention. If full attention becomes a bottleneck at extreme lengths, it can be replaced by sparse attention with a larger receptive field (e.g., sparse attention with RF=32K instead of full attention, paired with a sparser stream of RF=1K), preserving the bidirectional alignment philosophy while maintaining efficiency. We leave this exploration to future work.
> ### 3. Scale of the Experiment
> We conducted two preliminary experiments addressing both dimensions.
>
> **Model size (7B):** We trained a 7B model for 10B tokens. SSA achieves a lower training loss than the 1B model at the same token count (2.80 vs. 2.99), with no instability from the auxiliary loss, suggesting SSA scales faithfully with model size.
> | Model Size | Training Loss @10B tokens |
> |-|-|
> | 1B | 2.99 |
> | 7B | **2.80** |
>
> **Context length (128k):** We continually trained the 32k SSA model with 1B additional tokens at 128k context. PPL@128k improves under both inference modes (full: 13.84→13.65; sparse: 14.45→14.34), confirming SSA extends to longer contexts.
>
> | Context | PPL@128k (Full) | PPL@128k (Sparse) |
> |-|-|-|
> | 32k-continual-trained | 13.84 | 14.45 |
> | 128k-continual-trained | **13.65** | **14.34** |
>
>
> We acknowledge the scale limitation but note that 1B/100B-tokens is the established standard for validating efficient attention mechanisms, adopted by GLA [1], RAT [2], Multi-Token Attention [3], and others. Each additional experiment costs ~1,000 GPU hours, making further scaling prohibitive under our budget. This ablation-then-scale methodology is standard practice: SmolLM3 [4] and Kimi-K2 [5] both verified architectural decisions at comparable scales before full training runs. We leave large-scale validation to future work.
>
>
> [1] Gated Linear Attention Transformers with Hardware-Efficient Training.
>
> [2] RAT: Bridging RNN Efficiency and Attention Accuracy in Language Modeling.
>
> [3] Multi-Token Attention.
>
> [4] The Smol Training Playbook: The Secrets to Building World-Class LLMs.
>
> [5] Kimi K2: Open Agentic Intelligence.

---

> > ### Author Rebuttal · Reviewer_hxyn · 2026-04-01
> >
> > Thank you for the clarifications. My concerns are addressed. I'll keep my score.

---

> > > ### Author Response · Authors · 2026-04-01
> > >
> > > We sincerely thank Reviewer hxyn for the thorough review, which motivated us to add the complete RULER evaluation, 7B scaling experiments, and 128k context extension results — all of which we believe meaningfully strengthen the paper. We are glad our rebuttal addressed all concerns and appreciate the reviewer's support.

---

### Official Review · Reviewer_H1tJ · 2026-03-11

**Soundness:** 4
**Presentation:** 3
**Significance:** 4
**Originality:** 4
**Overall Recommendation:** 6
**Confidence:** 4

**Summary:**

The paper addresses two challenges (attention gap & capability gap) in sparse-attention training by introducing SSA, a dual-stream framework that combines sparse and full attention through attention-output alignment.  The method is well-motivated.
The paper provides extensive experiments and a clear analysis. The experimental results can demonstrate the superiority of the proposed method.

**Compliance With Llm Reviewing Policy:**

Affirmed.

**Final Justification:**

The rebuttal clarifies my concern. I confirm my original score.

**Key Questions For Authors:**

see **Weakness**

**Limitations:**

yes

**Strengths And Weaknesses:**

**Strengths**

The paper provides a very comprehensive evaluation, as well as solid theoretical proofs.
The authors report results on commonsense reasoning, perplexity, LongBench, and NIAH, and also analyze robustness under different sparsity budgets. Overall, SSA appears particularly strong under sparse inference and shows better performance across sparsity settings than several baselines.
These results support the paper’s main claim that SSA provides a good balance between sparse-inference adaptation and capability preservation.


**Weakness**

While Table 3 suggests that bidirectional alignment is important for training stability, it remains unclear whether the collapse of the one-way variants is intrinsic to the design itself or partly due to the choice of alignment-loss weight.

---

> ### Author Rebuttal · Authors · 2026-03-30
>
> We thank the reviewer for the detailed and constructive feedback.
> ### 1. Bidirectional Alignment
> **Can uni-directional alignment loss’s instability come from untuned hyperparameters?**
>
> This is an insightful point. We tuned α (the alignment loss coefficient) across [5, 20] as shown in Table A14, but all unidirectional variants exhibited exploding cross-entropy loss within a few hundred steps regardless of α. This consistent failure across multiple hyperparameter settings suggests the instability is intrinsic to the unidirectional design rather than an artifact of hyperparameter choice.

---

> > ### Author Rebuttal · Reviewer_H1tJ · 2026-04-01
> >
> > The rebuttal clarifies my concern. I confirm my original score.

---

> > > ### Author Response · Authors · 2026-04-01
> > >
> > > We sincerely thank Reviewer H1tJ for the constructive feedback. The question on unidirectional alignment instability vs. hyperparameter tuning helped us provide stronger empirical evidence for our design choices. We are glad that our rebuttal fully addressed the concerns and appreciate the reviewer's support.

---

### Official Review · Reviewer_wsDt · 2026-03-12

**Soundness:** 3
**Presentation:** 3
**Significance:** 3
**Originality:** 3
**Overall Recommendation:** 4
**Confidence:** 4

**Summary:**

This paper proposes SSA, a unified sparse attention training framework that aims to address two problems: the distribution mismatch when full-attention-trained models are used with sparse inference, and the performance degradation of purely sparse-trained models due to incomplete gradient flow. At each training step, SSA selects full attention or sparse attention streams with equal probability and encourages sparser attention distributions through a bidirectional alignment loss. The authors theoretically prove that the approximation error scales linearly with the dropped attention mass, and validate SSA's superiority on various  benchmarks.

**Compliance With Llm Reviewing Policy:**

Affirmed.

**Final Justification:**

The authors conducted additional experimental validations in the rebuttal (including under larger parameter settings), which enhanced the paper's reliability and addressed my concerns. I maintain my positive score.

**Key Questions For Authors:**

1. How does SSA perform on 7B or larger models? Does the training overhead scale with model size? Are there any preliminary scaling experiments?

2. The alignment loss in SSA is inherently similar to self-distillation. Could the authors more explicitly discuss the distinctions and connections between SSA and traditional knowledge distillation methods? Have you experimented with using a separate full-attention teacher model instead of computing the counterpart attention online?

**Limitations:**

yes

**Strengths And Weaknesses:**

**Strengths**

1. The paper clearly decomposes the challenges of sparse attention into two complementary problems—the "attention gap" and the "capability gap." The theoretical framework provides a solid motivation for the method design, ensuring that each design choice in SSA is grounded in theory.

2. The bidirectional alignment mechanism with sparsity loss and commitment loss is intuitively compelling—it encourages the full attention distribution to become inherently sparser while preventing sparse attention from drifting away from the representational space of full attention. The ablation studies further confirm that unidirectional alignment leads to training instability, underscoring the necessity of the bidirectional design.

3. The evaluation covers multiple dimensions including perplexity, commonsense reasoning, and long-context understanding, with systematic comparisons across different sparsity budgets. SSA outperforms or matches all baselines in nearly every setting, demonstrating strong flexibility under both sparse and full attention inference modes.

**Weaknesses**

1. All experiments are conducted on 300M and 1B models with 50B–100B training tokens. Given the current landscape of LLM research, this scale is insufficient to convincingly extrapolate SSA's effectiveness to 7B or even larger models. Since the practical value of sparse attention lies primarily in large-scale long-context scenarios, the lack of large-scale validation is a notable limitation.

2. Table A14 shows that the choice of α has a noticeable impact on performance, yet the paper provides no systematic guidance for hyperparameter selection. It remains unclear whether the optimal α needs to be re-tuned across different model scales and datasets.

3. On the NIAH task beyond 8k context length, SSA's performance drops sharply, with accuracy falling to single-digit percentages at 32k. The authors attribute this to being outside the alignment training distribution, but this is precisely the scenario where sparse attention is most needed. This limitation undermines the practical value of the method.

---

> ### Author Rebuttal · Authors · 2026-03-30
>
> We thank the reviewer for the detailed and constructive feedback.
>
> ### 1. Scale of the Experiment
> We conducted two preliminary experiments addressing both dimensions.
>
> **Model size (7B):** We trained a 7B model for 10B tokens. SSA achieves a lower training loss than the 1B model at the same token count (2.80 vs. 2.99), with no instability from the auxiliary loss, suggesting SSA scales faithfully with model size.
> | Model Size | Training Loss @10B tokens |
> |-|-|
> | 1B | 2.99 |
> | 7B | **2.80** |
>
> **Context length (128k):** We continually trained the 32k SSA model with 1B additional tokens at 128k context. PPL@128k improves under both inference modes (full: 13.84→13.65; sparse: 14.45→14.34), confirming SSA extends to longer contexts.
>
> | Context | PPL@128k (Full) | PPL@128k (Sparse) |
> |-|-|-|
> | 32k-continual-trained | 13.84 | 14.45 |
> | 128k-continual-trained | **13.65** | **14.34** |
>
>
> We acknowledge the scale limitation but note that 1B/100B-tokens is the established standard for validating efficient attention mechanisms, adopted by GLA [1], RAT [2], Multi-Token Attention [3], and others. Each additional experiment costs ~1,000 GPU hours, making further scaling prohibitive under our budget. This ablation-then-scale methodology is standard practice: SmolLM3 [4] and Kimi-K2 [5] both verified architectural decisions at comparable scales before full training runs. We leave large-scale validation to future work.
>
> ### 2. Alpha Tuning
> The guidance for alpha tuning is to balance the scale of the two loss terms: the cross-entropy loss operates in probabilistic space while the alignment loss operates in Euclidean space, and their magnitudes differ significantly. In our 1B setting, the initial cross-entropy loss is ~12 while the alignment loss is 40x smaller; we set \alpha=10 to bring the alignment loss to a comparable scale of influence. For the 7B model, the initial alignment loss increases by ~2.5x, so we decrease $\alpha$  to 4, and training remains stable. This yields a practical and principled tuning rule: set $\alpha$ so that the two loss terms are within roughly the same order of magnitude. We can follow standard hyperparameter search practice: training on 1-5% of data and selecting the value that minimizes validation loss.
> ### 3. Performance drop in NIAH beyond 8k
> This limitation is shared by all methods—FullAttn and MoBA also drop to single-digit accuracy at 32k under sparse inference (Table 3). To provide a more comprehensive evaluation, we evaluated all 8 NIAH variants from RULER @32k as shown in the table below. SSA consistently outperforms both baselines by a large margin under both inference modes:
>
> | Inference Mode | Full/% | Sparse/% |
> |-|-|-|
> | SSA | 30.34 | 14.14 |
> | MoBA | 3.05 | 1.34 |
> | FullAttn | 22.79 | 5.43 |
>
> SSA's advantage is substantial (2.6× over FullAttn, 10.5× over MoBA under sparse mode), demonstrating that SSA remains the strongest method even in this challenging regime. We will include full RULER results in the revision.
> ### Response to Key Questions
> **7B Model Training:** SSA scales faithfully to 7B as discussed above. Notably, SSA's training overhead decreases with model size—only ~8% longer than MoBA at 7B (vs. ~17% at 1B), with comparable memory usage. This is because attention becomes a smaller fraction of total compute at larger scales (FFN and projections dominate), so the auxiliary attention pass contributes proportionally less overhead. FlashAttention's efficiency further keeps the absolute cost low.
>
> **On connection to distillation:** SSA shares similarities with self-distillation but differs in a key way: conventional distillation is unidirectional (teacher→student), whereas SSA performs bidirectional alignment. Full attention learns sparser features from the sparse stream (Eq. 6), while sparse attention is regularized toward full attention's representations (Eq. 7). This bidirectionality is essential—the two streams address complementary problems (capability gap vs. attention gap), and removing either direction causes training collapse (Table 2, NaN).
>
> **On using a separate teacher:** This would require first pretraining a full-attention teacher, then training the student—doubling total pretraining cost. In the current scaling era where large models are trained only once under tight compute budgets, this is impractical. SSA instead co-trains both streams in a single run with only ~17% overhead over MoBA. Furthermore, a frozen teacher cannot benefit from the sparse stream's signal to learn inherently sparser distributions—a core mechanism behind SSA's superior attention sparsity.
>
> [1] Gated Linear Attention Transformers with Hardware-Efficient Training.
>
> [2] RAT: Bridging RNN Efficiency and Attention Accuracy in Language Modeling.
>
> [3] Multi-Token Attention.
>
> [4] The Smol Training Playbook: The Secrets to Building World-Class LLMs.
>
> [5] Kimi K2: Open Agentic Intelligence.

---

> > ### Author Rebuttal · Reviewer_wsDt · 2026-04-03
> >
> > Thank you for your clarifications. My concerns are addressed.

---

> > > ### Author Response · Authors · 2026-04-03
> > >
> > > We sincerely thank Reviewer wsDt for the constructive feedback. Your questions on scaling behavior, hyperparameter tuning, NIAH, and the connection to self-distillation helped us strengthen the paper significantly. We are glad that our rebuttal has addressed the concerns, and the corresponding improvements will be incorporated into the revised manuscript. We hope the revised version better reflects the contributions of this work.

---

### Official Review · Reviewer_hmKu · 2026-03-13

**Soundness:** 2
**Presentation:** 2
**Significance:** 1
**Originality:** 1
**Overall Recommendation:** 2
**Confidence:** 4

**Summary:**

This paper proposes a training strategy for sparse approximations of softmax attention that aims to narrow the gap between sparse and dense attention in transformers. The paper categorizes attention methods along two dimensions: whether training uses full or sparse attention, and whether inference uses full or sparse attention. The proposed method, SSA, alternates between full and sparse attention during training with 50% chance and adds an alignment loss so that the two branches produce similar hidden representations. One term encourages the full-attention branch to move toward the sparse branch, while the other keeps the sparse branch close to the full-attention branch in the L2 sense. Experiments on a 1B-parameter transformer trained on 100B tokens show modest gains over baselines on commonsense benchmarks, somewhat larger gains in language-model perplexity with sparse inference, and more pronounced improvements on long-context benchmarks, especially NIAH of RULER.

**Compliance With Llm Reviewing Policy:**

Affirmed.

**Final Justification:**

The authors made honest efforts to improve their empirical claims. However, I believe the mathematical parts of the paper need a major revision. In its current shape, I think this paper needs more mathematical rigor, therefore I maintain my score.

**Key Questions For Authors:**

Do you have results for different values of \alpha?
What are the efficiency aspects of SSA?

**Limitations:**

No. See the weakness points where I gave some suggestions.

**Strengths And Weaknesses:**

Pros:
- The paper is well-written and goes direct to the point. Figure 2 is great.
- The addressed problem is relevant: how to make sparse approximations of dense attention work well without training only with dense patterns.
- The empirical setup is reasonably strong: 1B model trained on 100B tokens; results with both full and sparse inference; evaluations on short- and long-context benchmarks; standard deviations are shown.
- The idea of using dense signals to guide sparsity selection is interesting and impactful

Cons:
- The paper overstates the weakness of fully sparse methods and misses important related work, like AdaSplash [1]

- Novelty of the method is somewhat moderate: the sparse selection mechanism itself feels close to prior block-sparse approaches which all rely on top-k for sparsification and mean-pooling for block-wise sparsity.

- The method still depends on full attention during training, which makes the overall efficiency aspect somewhat bad

- While I liked the alignment loss, I believe its optimization should be very delicate. In fact, the paper reports NaN results in the ablation table. I would not be surprised to see a significant impact in choosing $\alpha$

- Some theoretical claims are not fully convincing, and parts of the analysis seem fairly straightforward.
   - Proposition 1 is not fully convincing: the claim that zero gradients on dropped connections imply a "learning deficiency" is somewhat contradictory to what InfLLMv2 and AdaSplash report. I could argue that zero gradient can also indicate that those connections are unnecessary.
   - Also in Proposition 1, the statement that sparse masking prevents the model from learning to attend to tokens outside the selected block is too strong. Such connections can still be learned indirectly across many optimization steps (the previous weights of the network can be adjusted so that an "inactive" connection becomes "active".

  - The second part of Proposition 1 (“Attention Suppression Deficiency”) was unclear to me and not sufficiently justified.

- Notation and writing need improvement; some expressions and definitions are unclear.
  - - What is $(:t)$? I assume it's Python notation, but does it include $t$ or not?
  - Equation 2 may not be fully well defined as written, since the KL term can involve division by zero when sparse probabilities contain zeros.

- Theorem 4.5 is mathematically quite straightforward. It essentially follows from a simple norm inequality plus the dropped-mass term, so its insight appears limited to me.

- In the evaluation side, I've noticed some shortcomings as well
  - The gains on the main benchmark table are quite modest, with SSA often performing similarly to baselines, especially when considering std values.
   - Figure 3 uses receptive field as the x-axis, but for this paper sparsity ratio and actual efficiency metrics would be more informative. Overall, I think the evaluation part of the paper would be much stronger with explicit efficiency benchmarks.
   - The long-context evaluation also feels incomplete. For example, I would have liked to see more RULER tasks beyond NIAH.
   - Some key comparisons are missing, such as NSA numbers in Figure 4.


[1] Gonçalves, Nuno, Marcos Treviso, and André FT Martins. "Adasplash: Adaptive sparse flash attention." ICML 2025

---

> ### Author Rebuttal · Authors · 2026-03-30
>
> We thank the reviewer for the detailed and constructive feedback.
> ### 1. Related Work
> **Overstating the weakness of sparse methods:** We do not claim sparse attention is inherently ineffective, but rather that training exclusively with sparse or full attention is suboptimal for sparse inference performance.
>
> **AdaSplash [1,2]:** AdaSplash interpolates between Softmax and Sparsemax at the architecture level, while SSA achieves the same goal at the training-strategy level. We attempted to pretrain AdaSplash (α=1.5) under our setting (1B model, 100B tokens, 8k context), but training diverged to NaN because attention weights collapsed to all zeros. As shown in the table below, the instability arises due to the increased model size: we find its training is stable for hundreds of steps at AdaSplash's original scale (124M) but not at 1B, even after tuning a range of α (1.2-1.5), whereas SSA remains stable at this scale.
> |Model Size|Alpha|Training Loss@200 steps|First Failed Step|Initial Zero-Attn-Score-Ratio|
> |-|-|-|-|-|
> |1.2B|1.5|NaN|10|0.992|
> |1.2B|1.2|NaN|8|0.574|
> |730M|1.5|NaN|10|0.992|
> |370M|1.5|5.95|N/A|0.977|
> |100M|1.5|6.01| N/A|0.930|
> ### 2. Novelty of the Sparse Mechanism
> Our contribution is not a new sparse mechanism but a training framework that jointly optimizes full and sparse attention for better sparse inference. SSA is sparse-mechanism-agnostic—we use MoBA-like attention, but any block-sparse or top-k variant is a drop-in replacement. The novelty is in how sparse attention is trained, not which sparse pattern is used.
> ### 3.  Efficiency of Adopting Full Attention
> This is a worthwhile tradeoff: SSA incurs only 17% training overhead over MoBA (Table A2), while enabling pure sparse attention at inference, where efficiency matters most. SSA achieves better performance than sparse-only training while preserving inference efficiency.
> ### 4. Alpha
> The NaN in our ablation (Table 3) occurs only when the alignment loss is removed entirely, not a symptom of tuning sensitivity. When the loss is present, performance remains stable across a range of values (Table A14), showing that SSA is robust to this hyperparameter.
> ### 5. Proposition 4.1
> **Zero gradients as learning deficiency:** We agree some dropped tokens may be irrelevant. However, at RF=256 with 8k context, each query observes only 6% of tokens per layer—assuming the remaining 94% are uninformative is a strong assumption. InfLLM-v2's [3] larger RF (~6k) mitigates this, which supports rather than contradicts our claim.
>
> **Indirect learning across steps:** We agree that the statement was too strong. Our intended claim is that the learnable information per step is reduced, not that specific tokens can never be attended to—different token positions and layers can attend to different blocks, allowing indirect coverage.
>
> **Attention Suppression Deficiency:** Excluding low-ranked tokens from the softmax denominator removes competitive pressure that would suppress irrelevant tokens to near-zero probability. This manifests as higher entropy in MoBA relative to FullAttn (Figure 1c).
> ### 6. Notations
> (:t) is inclusive of t, and the KL in Equation 2 is over vocabulary distributions, not attention weights—so all probabilities are strictly positive and the divergence is well-defined.
> ### 7. Theorem 4.5
> We acknowledge that the proof is technically straightforward. However, the value of Theorem 4.5 lies in **prescriptive insight**: it formally identifies δ(t) as the key quantity governing the performance gap between sparse and full attention, and directly motivates SSA's bidirectional alignment design to minimize δ(t). Empirically, this translates to SSA achieving the smallest perplexity gap between sparse and full inference (0.68 vs. FullAttn's 1.92), validating the theorem's design guidance.
> ### 8. Evaluations
> **Modest Gains:**  Most commonsense tasks have context below 1024 tokens, so at RF=1024 sparse attention degenerates to near-full attention. At RF=256—the practically relevant setting—SSA outperforms FullAttn and MoBA by a clear margin (60.27% vs. 59.62% vs. 58.73%).
>
> **Efficiency Benchmarks:** SSA is a training framework, not a new sparse kernel. At inference, SSA uses the same sparse attention operator as MoBA—our contribution is improved output quality under the same sparsity constraint (Appendix D).
>
> **RULER:** We ran full RULER @32k (RF=256 extrapolated). SSA consistently outperforms baselines in both modes:
> |Inference Mode|Full/%|Sparse/%|
> |-|-|-|
> |SSA|25.48|12.63|
> |MoBA|2.57|3.61|
> |FullAttn|17.71| 5.69|
>
> **Missing NSA in Figure 4:** NSA's perplexity is in Table A8; we will add a cross-reference from Figure 4. NSA includes a sliding window module that favors perplexity—removing it makes NSA worse than SSA (Appendix G), so the comparison without this component is the fairer one.
>
> [1] AdaSplash: Adaptive Sparse Flash Attention
>
> [2] Sparse Sequence-to-Sequence Models
>
> [3] InfLLM-V2: Dense-Sparse Switchable Attention for Seamless Short-to-Long Adaptation

---

> > ### Author Rebuttal · Reviewer_hmKu · 2026-04-01
> >
> > I chose option (c) because my concerns are about the core theoretical framing and motivation of the paper, and I think the short rebuttal was not enough to clarify my concerns.
> >
> > I already wrote some comments above but I will try to clarify my main concerns further in here:
> >
> > 1. The overstatements about sparse attention weakness in the introduction (used as motivation) and then again in Proposition 4.1.
> >
> > 2. Proposition 4.1 lacks mathematical rigor. In fact, I am not convinced the proposition is valid at all.
> > - "gradient is zero"only implies gradient through a specific (i, j) edge at a specific training step. It does not imply the stronger statement "the model cannot learn to attend to tokens outside the selected ones". The authors then claim that what they mean is that "learnable information per step is reduced". But that is a much weaker thing than what is stated in the Proposition.
> > - The paper treats zero gradient as a deficiency; I don't think that is automatically true. It can simply mean that the connection is not needed or that the model is learning an useful inductive sparse structure.
> > - The "Attention Suppression Deficiency" is a bit unclear to me without a formal definition. Maybe it can serve as a reasonable intuition, but it's not a proved proposition.
> >
> > 3. I also remain concerned about the level of mathematical precision in Proposition 4.2. It follows from the direct (one step) definition of KL. The proof in the appendix uses an expectation $\mathbb{E}_{FA}$ but never clearly defines whether that's over tokens sampled from the full model, or the data distribution, or over contexts...
> >
> > 4. Theorem 5 actually looks sound to me. The mathematical steps in the appendix check out. My concern here is not with the theorem itself, but with the subsequent claim that the alignment objective directly reduces the dropped attention mass $\delta(t)$ (as stated in Section 5 after Eq. 6). What the theorem shows is that the attention output difference is bounded in terms of $\delta(t)$, and so it does not establish that minimizing $L_{sparsity}$ necessarily decreases $\delta(t)$. Perhaps more important, I don't think the claim is supported and seems generally false (correct me if I'm wrong). So, the paper defines
> >
> > $
> > L_{sparsity} = |h_{full}- sg(h_{sparse})|,
> > $
> >
> > and then says this objective "directly reduces the dropped attention mass $\delta(t)$".
> > That claim is not generally true, because $L_{sparsity}$ acts on the output vector $h_{full}$, while $\delta(t)$ is defined from the full-attention mass on dropped tokens:
> >
> > $
> > \delta(t)=\sum_{j\in S^c(t)} a_j^{full}(t).
> > $
> >
> > Those are different objects. A small output mismatch does not necessarily force small dropped attention mass.
> >
> > So, since these concerns essentially affect the theoretical core of the paper and the way the method is motivated, I don't think they can be fully resolved within a short author response alone. I think that addressing them would require substantial revision of the theoretical section (especially for mathematical rigor), a more thorough related work investigation to help with the overstated claims.

---

> > > ### Author Response · Authors · 2026-04-04
> > >
> > > We thank the reviewer and address each concern below.
> > >
> > > **On Proposition 4.1 (Concerns 1 & 2)**
> > >
> > > **Gradient scope.** Proposition 4.1 describes a *local* gradient property, not permanent learning inability. At each layer, each query position independently selects its own top-k blocks, and tokens outside those blocks receive zero gradient. This is a mathematical fact about block-sparse backward pass. Since different query positions at different layers select different blocks, the model can still learn about any token. Our point is that at any single query position in any single layer, the learning signal is incomplete compared to full attention.
> > >
> > > **Expressiveness vs. Regularization.** The reviewer suggests zero gradient may serve as a useful inductive bias (regularization). We find our views are not contradictory: zero gradient reduces expressiveness while potentially acting as regularization. However, the balance depends on scale. In our setting with 96% sparsity (RF=256) or 75% sparsity (RF=1024), the expressiveness loss dominates the regularization benefit. Table 1 supports this: MoBA under sparse inference underperforms FullAttn under full inference, suggesting that forced sparsity's regularization is outweighed by the limited token interactions available during training and inference.
> > >
> > > **Attention Suppression Deficiency.** Since softmax attention sums to 1, all tokens compete for a fixed budget of attention mass. Boosting any token's score automatically suppresses every other token. In sparse attention, excluded tokens are removed from the softmax entirely, so this competitive pressure disappears for them. As a result, sparse-trained models suppress irrelevant tokens more weakly than full-attention-trained ones, leading to higher scores on unrelated keys.
> > >
> > >
> > > **On Proposition 4.2 (Concern 3)**
> > >
> > > The reviewer correctly identifies a notational inconsistency in our proof: the subscript of the expectation was used inconsistently, sometimes over the full model's output distribution and sometimes over the data distribution.
> > >
> > > The fix is to consistently take the expectation over the full model's output distribution throughout. Under this definition, the proof becomes the standard KL identity:
> > >
> > > $$H(p^{\text{full}}, p^{\text{sparse}}) = H(p^{\text{full}}) + D_{\text{KL}}(p^{\text{full}} \| p^{\text{sparse}})$$
> > > where the two distributions are over the vocabulary at position $t$ conditioned on the same context.
> > >
> > > Importantly, this fix does not change the conclusion of Proposition 4.2: the KL divergence penalty from training-inference mismatch still explains why a full-attention-trained model can have *worse* sparse-inference loss than a sparse-attention-trained model (Remark 4.3). We will correct this in the revision.
> > >
> > > **On Theorem 4.5 and the Sparsity Loss (Concern 4)**
> > >
> > > The reviewer correctly notes that the sparsity loss operates on output vectors, while $\delta(t)$ is defined on attention weights. These are mathematically different objects. We agree that small output difference does not *formally guarantee* small $\delta(t)$, and we will revise the paper to avoid stating this as a direct mathematical implication.
> > >
> > > However, this is an intentional design choice. We chose output-level alignment because direct attention-weight alignment requires materializing dense $N \times N$ attention maps, incompatible with FlashAttention-2 [1] and incurring significant memory and speed overhead. Output-level alignment only compares hidden-state vectors, which is computationally cheap.
> > >
> > > We argue that output-level alignment is a good proxy for reducing $\delta(t)$. The reviewer's concern holds if value vectors are highly correlated, as correlated values could "cancel out" the dropped contribution without reducing $\delta(t)$. But we measured the mean pairwise cosine similarity of value vectors across layers to be only **0.1259** (*updated for more data @8k length*), indicating near-orthogonality. With approximately orthogonal values, the only way to make $h_{\text{full}}$ close to $h_{\text{sparse}}$ is to shrink attention mass on dropped tokens, i.e., reduce $\delta(t)$, since orthogonal contributions cannot cancel out. We further computed the Spearman rank correlation between $\|h_{\text{full}} - h_{\text{sparse}}\|$ and $\delta(t)$ across layers on held-out data:
> > >
> > > | Layer | 0 | 1 | 2 | 3 | 4 | 5 | 6 | 7 | 8 | 9 | 10 | 11 | 12 | 13 | 14 | 15 | Mean |
> > > |-|-|-|-|-|-|-|-|-|-|-|-|-|-|-|-|-|-|
> > > | Spearman | 0.45 | 0.86 | 0.76 | 0.82 | 0.85 | 0.65 | 0.56 | 0.63 | 0.82 | 0.89 | 0.71 | 0.93 | 0.86 | 0.82 | 0.89 | 0.80 | **0.77** |
> > >
> > > The mean correlation (0.77) confirms that reducing output difference reliably co-occurs with reducing dropped attention mass. Figure 1(c-d) further support this.
> > >
> > > We thank the reviewer for the constructive feedback. We will revise our paper accordingly, and include necessary related work such as AdaSplash.
> > >
> > > [1]: Tri Dao. FlashAttention-2: Faster attention with better parallelism and work partitioning. ICLR 2024.

---

### Decision · Program_Chairs · 2026-04-30

**Decision:**

Accept (regular)

**Comment:**

This paper proposes a training strategy for sparse approximations of softmax attention. The paper categorizes attention methods along two dimensions: whether training uses full or sparse attention, and whether inference uses full or sparse attention. The proposed method, Sparse Sparse Attention (SSA), alternates between full and sparse attention during training adding an alignment loss so that the two branches produce similar hidden representations. Experiments on a 1B-parameter transformer trained on 100B tokens show modest gains over baselines on commonsense benchmarks, somewhat larger gains in language-model perplexity with sparse inference, and more pronounced improvements on long-context benchmarks, especially NIAH of RULER.

There was significant disagreement among reviewers about this paper. Reviewers pointed out as strengths the effectiveness of the proposed method, the strong empirical performance of the proposed method, and the clear writing. However, they pointed out as weaknesses the poor characterization of fully sparse methods (where relevant related work is missing or their weakness exaggerated), the limited novelty of the proposed method, and the lack of rigor in some of the theoretical claims (particularly in Proposition 4.1). The rebuttal alleviated some of these concerns but not all of them.

During the discussion among reviewers, no full consensus was reached, however all reviewers agree that some theoretical statements should be revised for precision (e.g. related to Proposition 4.1) as well as the discussion about fully sparse methods. Most reviewers view the paper as a technically solid contribution nevertheless.

Overall, I believe this paper is worthy of acceptance at ICML if the issues above are addressed in the camera ready. I urge the authors to take the comments from the reviewers seriously, by fixing the issues in Proposition 4.1 and improving their characterization of fully sparse methods, properly acknowledging and discussing prior work in an objective manner.